# Calcium wave dynamics in the embryonic mouse gut mesenchyme: impact on smooth muscle differentiation
Nicolas R. Chevalier [1] ✉, Léna Zig[1], Anthony Gomis[1], Richard J. Amedzrovi Agbesi[1], Amira El Merhie[1],
Laetitia Pontoizeau[2], Isabelle Le Parco[3], Nathalie Rouach [4], Isabelle Arnoux [4],
Pascal de Santa Barbara[5] & Sandrine Faure [5]

Intestinal smooth muscle differentiation is a complex physico-biological process involving several different pathways. Here, we investigate the properties of $Ca^{2+}$ waves in the developing intestinal mesenchyme using GCamp6f expressing mouse embryos and investigate their relationship with smooth muscle differentiation. We find that $Ca^{2+}$ waves are absent in the pre-differentiation mesenchyme and start propagating immediately following α-SMA expression. $Ca^{2+}$ waves are abrogated by $Ca_V1.2$ and gap-junction blockers, but are independent of the Rho pathway. The myosine light-chain kinase inhibitor ML-7 strongly disorganized or abolished $Ca^{2+}$ waves, showing that perturbation of the contractile machinery at the myosine level also affected the upstream $Ca^{2+}$ handling chain. Inhibiting $Ca^{2+}$ waves and contractility with $Ca_V1.2$ blockers did not perturb circular smooth muscle differentiation at early stages. At later stages, $Ca_V1.2$ blockers abolished intestinal elongation and differentiation of the longitudinal smooth muscle, leading instead to the emergence of KIT-expressing interstitial cells of Cajal at the gut periphery. $Ca_V1.2$ blockers also drove apoptosis of already differentiated, $Ca_V1.2$-expressing smooth muscle and enteric neural cells. We provide fundamental new data on $Ca^{2+}$ waves in the developing murine gut and their relation to myogenesis in this organ.

Gut motility arises from the coordinated contractions of intestinal smooth muscle (SM) cells. Digestive SM cells derive from the embryonic mesoderm that gives rise to the mesenchyme, which in turn differentiates into submucosa, KIT-positive interstitial cells of Cajal (ICCs), and SM cells[1,2]. Digestive mesenchymal progenitors are defined by the expression of the LIX1 gene[3]. Their differentiation into SM requires the induction of *MYOCD*, a master regulator of SMC-restricted gene expression. Determined SM cells are defined by the early expression of the alpha and gamma isoforms of smooth muscle actin (respectively, αSMA and γSMA). Later during development, SM cells organize in layers and express proteins involved in contractility, such as CALPONIN and SM22. SM organization in the gut is largely conserved in species as phylogenetically distant as hydra and humans[4]. The SM is comprised of an inner circular smooth muscle (CSM) layer, an outer longitudinal smooth muscle (LSM) layer, and depending on species, a further thin circular and/or longitudinal layer

located between lamina propria and submucosa layers called the muscularis mucosae. The CSM and LSM are the essential effectors of bolus motion by peristalsis[5] or segmentation[6], while the muscularis mucosae drive smaller-scale motion of epithelial villi and of the lymph vessels circulating within them.

The most important signaling pathways involved in CSM differentiation are the sonic hedgehog (Shh) and bone morphogenetic protein pathways (BMP). Modulation of these pathways has been shown to result in strong alterations of CSM genesis, from complete disappearance[7,8] to hypertrophy[9,10] and misalignment[11,12]. The platelet-derived growth factor (PDGF) pathway is also critical because its inhibition leads to the disappearance of the LSM in mice[13].

It is now well understood that SM emergence depends not only on biochemical factors but also on mechanical tensile stress acting within the gut tissue: internal circumferential tension due to rapid proliferation of the

[1]Laboratoire Matière et Systèmes Complexes, Université Paris Cité, CNRS UMR 7057, 10 rue Alice Domon et Léonie Duquet, 75013 Paris, France. [2]ANIMAL-LIANCE, Insourcing Department, Paris, France. [3]Université Paris Cité, CNRS, Institut Jacques Monod, 75013 Paris, France. [4]Neuroglial Interactions in Cerebral Physiology and Pathologies, Center for Interdisciplinary Research in Biology, Collège de France, CNRS, INSERM, Labex Memolife, Université PSL, Paris, France. [5]PhyMedExp, University of Montpellier, INSERM, CNRS, Montpellier, France. ✉e-mail: nicolas.chevalier@u-paris.fr

epithelium causes the first CSM layer to orient circularly[9]. Spontaneous contractions of the CSM generate cyclic circumferential compression and longitudinal tension that orients the LSM longitudinally[14,15]: halting CSM contractions before the differentiation of the LSM was found to result in aberrant orientation of this second smooth muscle layer in the chicken embryo[9].

Recently, the bioelectrical events underlying the differentiation and morphogenesis of organs have attracted much attention[16]. It is becoming clear that the polarization state of the cell (membrane potential), the electric connectivity between groups of cells (gap junctions), and their ability to propagate coordinated signals like calcium waves (CWs), all play important roles in development. Thus, the epithelium of the wing of the *Drosophila melanogaster* embryo spontaneously generates CWs, and disruption of this pattern leads to abnormal wing development[17]. $Ca^{2+}$ oscillations have been detected in the developing chicken feather buds and found to play a critical role in their morphogenesis[18]. Chondrogenesis in the mouse limb bud has been shown[19] to be preceded by a depolarization event that induces $Ca^{2+}$ transients mediated by the L-type $Ca^{2+}$ channel $Ca_V1.2$; perturbations of the physiological electric pattern led to abnormal limb growth. We have shown previously that the contractile waves in the developing chicken intestine (at stage E9) are mediated by gap-junction dependent, mechanosensitive CWs that propagate in the smooth muscle syncytium immediately after it differentiates in the proximal midgut at E5[20,21]. More recently, propagation of CWs has been reported in the early chicken gut mesenchyme using GCaMP6S (calcium sensor) expression[9]. These different facts motivated us to examine the dynamics of $Ca^{2+}$ signals in the mouse intestinal mesenchyme before and after smooth muscle differentiation and to test whether early $Ca^{2+}$ waves could play a morphogenetic role in the emergence of this tissue layer.

To this end, we use here a genetically modified mouse line that expresses the fluorescent reporter GCamp6f in the mesenchyme. We first establish the detailed chronology of CSM and LSM differentiation in the mouse lower gastrointestinal tract, from duodenum to colon. We then examine the dynamics of $Ca^{2+}$ waves in relation to this chronology; we find that CSM $Ca^{2+}$ waves are present immediately after differentiation but not prior to differentiation and that the first detectable contractions induced by CWs appear one day after differentiation. We then pharmacologically characterize the properties of these early CWs and describe their evolution at different developmental stages. We next present experiments where $Ca^{2+}$ waves are inhibited by L-type $Ca^{2+}$ channel blockers at stages prior to CSM or LSM differentiation. We find that L-type channel blockers do not inhibit CSM differentiation, indicating that early $Ca^{2+}$ waves are not required for the differentiation of this layer. Application of $Ca_V1.2$ blockers after CSM differentiation and before LSM differentiation induced apoptosis of $Ca_V1.2$ expressing CSM cells and enteric neurons and completely inhibited the formation of the LSM, leading instead to the differentiation of KIT-expressing interstitial cells of Cajal at the gut periphery. We conclude by discussing possible mechanisms underlying our experimental results.

## Results
### Chronology of α-SMA and $Ca_V1.2$ expression in the mouse GI tract
We first characterized in detail by whole-mount (Fig. 1) and frozen slice (Fig. 2) immunohistochemistry (IHC) for α-smooth muscle actin (α-SMA) the chronology of smooth muscle differentiation in the proximal midgut (duodenum– proximal jejunum), distal midgut (distal jejunum–ileum) and colon. We distinguished SMA fibers on the following criteria: (1) a fibrous appearance of the staining, (2) its presence as two distinct layers on either side of the epithelium, and not extending all the way to the outer rim of the sample (serosa). Staining which did not fulfill these criteria was considered non-specific antibody labeling. Whole-mount staining ($n = 3$ embryos per stage) revealed no SM fibers at E11.5. The first fibers appeared in the proximal and distal midgut at stage E12.5—fibers in proximal portions were longer and more conspicuous. The first SM fibers in the colon appeared at stage E13.5. The first LSM fibers appeared in the proximal midgut at E15.5

but were absent in the more distal regions (ileum, colon). Frozen-section IHC results (Fig. 2) were consistent with whole-mount IHC (Fig. 1). We further labeled L-type $Ca^{2+}$ channels ($Ca_V1.2$) that, as we will show, are instrumental in the genesis and propagation of $Ca^{2+}$ waves in the developing smooth muscle. $Ca_V1.2$ signal in E11.5 and E12.5 midgut (Fig. 2) was diffusely expressed in the mesenchyme, the signal being most intense at the outer border of the samples and decreased as one traveled towards the tissue interior. The signal did not co-localize with either enteric neurons (E11.5–E12.5) or the smooth muscle layer (E12.5). The average $Ca_V1.2$ signal in the mesenchyme was most intense at E11.5 and decreased at later developmental stages (E12.5–E14.5, see inset top right in Fig. 2). We do not think this diffuse signal is due to non-specific binding at the tissue border because it was not present at the epithelial tissue border, nor in other holes (i.e., artificial borders) present in the tissue. As from E13.5, the $Ca_V1.2$ signal became restricted to the CSM layer (red arrow, Fig. 2) and to enteric neurons (white arrows, Fig. 2). Interestingly, $Ca_V1.2$ expression was more intense as from E13.5 in enteric neurons than in the surrounding smooth muscle.

### $Ca^{2+}$ dynamics in the intestinal mesenchyme during smooth muscle differentiation
We imaged $Ca^{2+}$ transients in the developing mesenchyme at stages E11.5 ($n = 3$), E12.5 ($n = 9$), E14.5 ($n = 4$), and in E14.5 guts cultured for 2 days (E14.5 + 2, $n = 4$) using a genetically modified mouse line which expressed the GCaMP6f reporter in the mesenchyme (see the "Methods" section). We did not detect any $Ca^{2+}$ wave (CW) activity in pre-differentiation E11.5 midgut ($n = 3$, Video S1) or in E12.5 colon ($n = 2$, Video S2). The first CWs became apparent immediately upon differentiation of SMC in the midgut at E12.5 (Fig. 3a, Video S1), distinctly outlining the circumferentially aligned cell bodies of the CSM. In $n = 6/9$ E12.5 samples, CWs did not induce a detectable contraction of the smooth musculature (Video S1), while 3/9 samples presented slight, barely visible contractions. At stage E14.5, midgut CSM CWs were accompanied in all samples by a distinct contraction of the CSM (Fig. 3b, Video S1), which was even more prominent at E14.5 + 2 (Fig. 3c, Video S1). CWs originated from random points along the gut[22]. After nucleation, the CW quickly propagated in the circumferential direction and more slowly along the longitudinal direction, giving rise to a rostro-caudal and a caudo-rostral wave (Fig. 3c). The frequency of CWs increased from ~3 cpm at stages E12.5–E14.5 to ~8 cpm at E14.5 + 2 (Fig. 3d). Interestingly, the first CWs in the midgut at E12.5 had the highest longitudinal propagation speed of ~360 μm/s, this velocity then significantly decreased at E14.5 to ~50 μm/s and finally increased again at E14.5 + 2 to ~100 μm/s (Fig. 3e).

Colonic CWs were absent at E12.5 ($n = 2$), but could be detected in all E14.5 ($n = 4$) and E14.5+ 2 ($n = 4$) preparations (Video S2). CW frequency and velocity were significantly lower in the colon than in the midgut at both E14.5 and E14.5 + 2 (Fig. 3e).

We could not detect any longitudinal contractions or CW associated to the LSM in the midgut, neither pre-differentiation (at E14.5), nor post-differentiation (at E14.5 + 2). The only visible $Ca^{2+}$ activity in the LSM at E14.5 + 2 consisted of unsynchronized blinking of longitudinally aligned cells at the gut periphery (Fig. 3c arrowhead, see Video S1).

### Molecular determinants of CW propagation
We next sought to understand the molecular underpinnings of CW generation in the murine gut. We performed pharmacological experiments at E12.5 in the midgut, i.e., right after CSM had differentiated. Nifedipine, an L-type calcium channel blocker, immediately abrogated CWs ($n = 4$, Fig. 4a, b, Video S3). Conversely, the $Ca_V1.2$ agonist (S)-(−)-Bay K8644 increased the amplitude and duration of $Ca^{2+}$ events immediately after application; this behavior was only transitory, however, and shifted 15 min after application to high-frequency, short duration, low-intensity CWs (Fig. 4c–e, Video S4). The effects of the gap-junction blocker enoxolone were similar to the one we previously reported in the chicken embryo[20]: enoxolone induced within 5 min after application a burst in the frequency of

| Proximal midgut | Distal midgut | Hindgut |
|---|---|---|

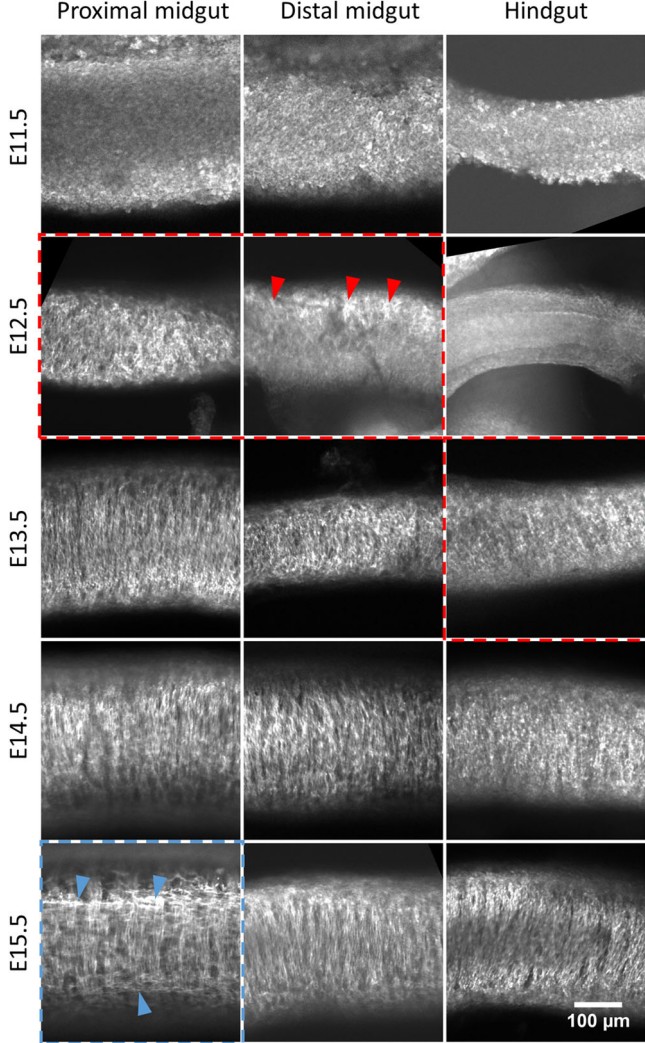

**Fig. 1 | Whole-mount staining for α-SMA at stages E11.5–E15.5 in the proximal midgut (duodenum–proximal jejunum), distal midgut (distal jejunum–ileum) and colon.** The first detectable CSM fibers (dashed red rectangles, red arrowheads) arise at E12.5 in the proximal and distal midgut, and at E13.5 in the colon. The first LSM fibers (dashed blue rectangle, blue arrowheads) appear in the proximal midgut at E15.5. The scale of all images is indicated in the bottom-right corner.

CWs; their amplitude then gradually decreased until no CWs were left after ~10 min (Fig. 4f, g, Video S5). This confirms the importance of gap junctions in this multi-cellular CW.

Myosine contractility is generally thought[23,24] to be downstream of $Ca^{2+}$-related events (calmodulin activation) and we expected that interfering with myosine would not affect upstream $Ca^{2+}$ events. Surprisingly, we found that ML-7, an inhibitor of myosine light-chain kinase (MLCK), profoundly altered upstream calcium handling: it significantly reduced CW frequency at 5 µM (Fig. 4h); at 10 and 50 µM, the synchronized CWs became uncoordinated. Whereas in controls, all cells along the diameter in the field of view would light up simultaneously, $n = 2/5$ samples at 10 µM and $n = 5/5$ at 50 µM showed an erratic propagation, with cells along the diameter not lightning up simultaneously (Fig. 4j, Video S6), and a strongly decreased propagation speed of $24 \pm 10$ µm/s ($n = 7$) compared to controls ($356 \pm 49$ µm/s). After 20 min in ML-7 10 and 50 µM, CWs vanished in, respectively, $n = 2/2$ guts and $n = 4/5$ guts (Fig. 4h, i).

We finally examined whether the Rho/ROCK pathway[23,24] is involved in CW propagation and smooth muscle contractility. For these experiments, because we lacked sufficient guts expressing GCaMP in the mesenchyme, we examined contractile waves at E12.5 + 2 guts (i.e. when distinct contractile

waves appear) instead of CWs at E12.5. Guts at E12.5 were cultured for 2 days either with DMSO vehicle alone (control, $n = 4$), or with the Rho/ROCK inhibitor Y-27632 at 1 µM ($n = 3$) or at 10 µM ($n = 3$). Contractile frequencies at E12.5 + 2 between these different groups were not significantly different (Fig. 4k, l, Video S7). The Y-27632 concentration of the 1 µM samples was then increased to 10 µM (first red arrow in Fig. 4l), the Y-27632 concentration of the 10 µM samples to 50 µM (second red arrow in Fig. 4l), and the effect on motility immediately monitored: we could not evidence any significant changes in the contractile wave frequency (Fig. 4l, Video S7). These results show that the Rho/ROCK pathway is not involved in early embryonic gut contractility, as had been assumed by other investigators[11].

## Calcium waves are independent of circular smooth muscle differentiation

These pharmacological experiments provided us with tools to modulate $Ca^{2+}$ activity in the developing gut and thus assess their importance for smooth muscle differentiation. We first investigated whether abrogating CWs with nifedipine would alter CSM differentiation, by culturing E11.5 guts for 2 days in the presence of this drug. Control guts after culture exhibited high CSM coverage of $88 \pm 22\%$ ($n = 10$) in the midgut (Fig. 5a, b). CSM differentiated along the whole length of the colon in 3/7 samples; in 1/7, just in the proximal colon; 3/7 samples did not display any CSM in the colon (Fig. 5b). Control guts exhibited phasic contractions at E11.5 + 2 in the midgut at a frequency of $1.9 \pm 0.3$ cpm ($n = 10$, Fig. 5c). We found that CSM normally differentiated in the presence of nifedipine 10 µM (Fig. 5a) and the resulting CSM coverage after 2 days of culture was not different from control samples in the midgut and colon (Fig. 5a, b). ENS morphology was similar in control and nifedipine-treated samples (Fig. 5a). Contractions were completely abolished in the guts cultured in nifedipine 10 µM, confirming that the compound had efficiently abrogated $Ca^{2+}$ activity (Fig. 5c).

Treatment of E11.5 guts with ML-7 5 µM, the minimum concentration from which this compound significantly reduced CW activity (Fig. 4i), resulted in organ death ($n = 3/3$, guts were small and opaque in transmitted light). We further performed experiments using BayK 8644 (2 µM, $n = 4$) to stimulate CW during culture (Fig. 4c). E11.5 + 2 guts treated with this compound did not display any visible differences in CSM differentiation in the midgut or colon compared to controls. These experiments suggest that early CSM CWs are dispensable for CSM differentiation and that modulating their activity does not affect the emergence or morphology of this first smooth muscle layer.

## Nicardipine abolishes longitudinal smooth muscle differentiation and promotes differentiation of interstitial cells of Cajal

We next examined how blocking $Ca_V1.2$ influenced the longitudinal smooth muscle layers by culturing E13.5 guts for 3 days (E13.5 + 3) in nicardipine or E14.5 guts for 2 days (E14.5 + 2) in nifedipine. The choice of nifedipine or nicardipine for a given set of experiments was a matter of convenience, both blockers having similar potency[25]. LSM differentiated in culture: E13.5 + 3 and E14.5 + 2 controls (i.e. DMSO vehicle only) displayed an LSM coverage of $25 \pm 17\%$ ($n = 6$) and $66 \pm 13\%$ ($n = 6$), respectively (Fig. 6a), as assessed from whole-mount IHC. Treatment with $Ca_V1.2$ blockers at 10 µM completely abolished LSM differentiation in E13.5 + 3 guts and significantly reduced it to $6 \pm 4\%$ in E14.5 + 2 samples (Fig. 6a). Nicardipine 10 µM also completely abolished motility at E13.5 + 3 (Fig. 6b). Treatments with lower concentrations of nicardipine (0.1, 1 µM) did not significantly reduce CSM coverage or motility at E13.5 + 3 (Fig. 6a, b). Interestingly, when pooling together the $n = 23$ samples from our nicardipine dose experiment at E13.5 + 3, we found a striking correlation between the presence of motility and the length increase of the intestines: samples displaying motility ($n = 11$) elongated (see the "Methods" section) while non-motile samples ($n = 12$) shrunk (Fig. 6c).

We further confirmed LSM coverage results obtained from whole-mount IHC by immuno-staining frozen sections of E14.5 + 2 jejunum (Fig. 6d, e, $n = 7$ controls, $n = 8$ nifedipine). Whereas control samples

**Fig. 2 | Frozen-section staining for Ca$_V$1.2, α-SMA, and βIII-tubulin (Tuj1) at stages E11.5–E14.5 in longitudinal sections of the midgut (jejunum).** All images were acquired at equal illumination intensity, exposure time, and contrast adjustment settings so that the intensity could be compared between different stages for each antibody applied. Ca$_V$1.2 is diffuse at E11.5 and E12.5 and becomes restricted to the SMC and enteric neurons as from E13.5. The inset in the top right corner shows that the average Ca$_V$1.2 signal intensity in the mesenchyme is higher at E11.5 than at later stages ($n = 8$ slices from $n = 2$ different embryos at each stage), *$p < 0.05$, Mann–Whitney two-tailed test. Red and white arrows at E13.5 and E14.5 point at Ca$_V$1.2 expression in the CSM and enteric neurons, respectively. Note that the position of the arrows is exactly the same for the three antibodies and the merged image. The CSM appears in the midgut at E12.5 (red dashed rectangle).

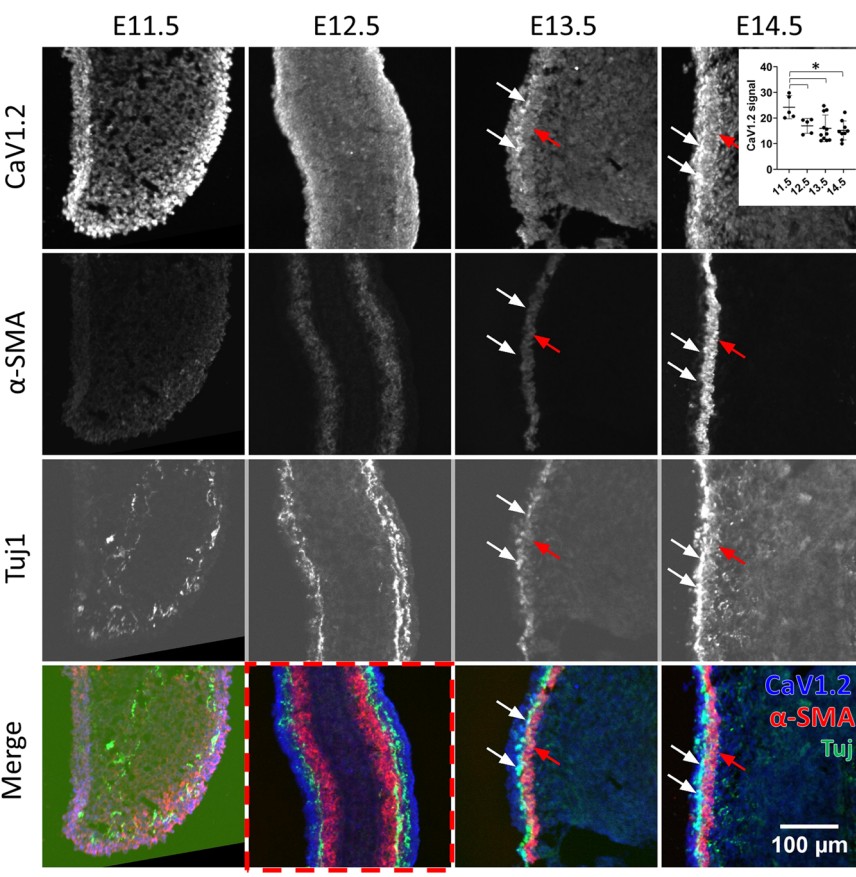

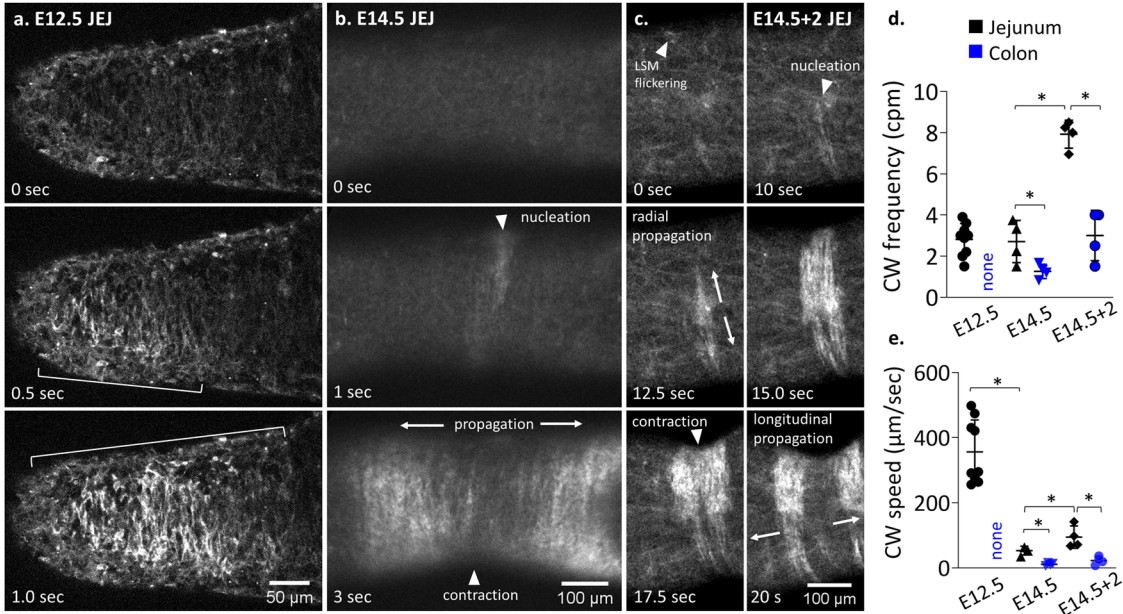

**Fig. 3 | Ca$^{2+}$ wave dynamics in the developing midgut mesenchyme.** At stages **a** E12.5, **b** E14.5 and **c** E14.5 + 2 and quantification of CW frequency (**d**) and longitudinal propagation speed (**e**) at these stages in the jejunum (black) and colon (blue). CWs were not present in the E12.5 colon. *$p < 0.05$, Mann–Whitney two-tailed test. Each dot represents a different sample.

displayed a well-differentiated LSM layer, this layer was absent in the nifedipine group. While the CSM layer was still present in nifedipine-treated samples, it was affected by the treatment, displaying a diminished density of fibers, with altered morphology (rough edges), and many rounded cells. Neuron density was decreased in nifedipine samples, and they presented

with scattered nerve fibers (Fig. 6d, e). These observations suggest apoptosis that we investigated by cleaved caspase-3 IHC. The fraction area of mesenchyme displaying cleaved caspase-3-positive cells was significantly higher in nifedipine than in controls (Fig. 6f). Apoptotic cells co-located both with CSM and ENS cells, but was absent from the outer gut periphery

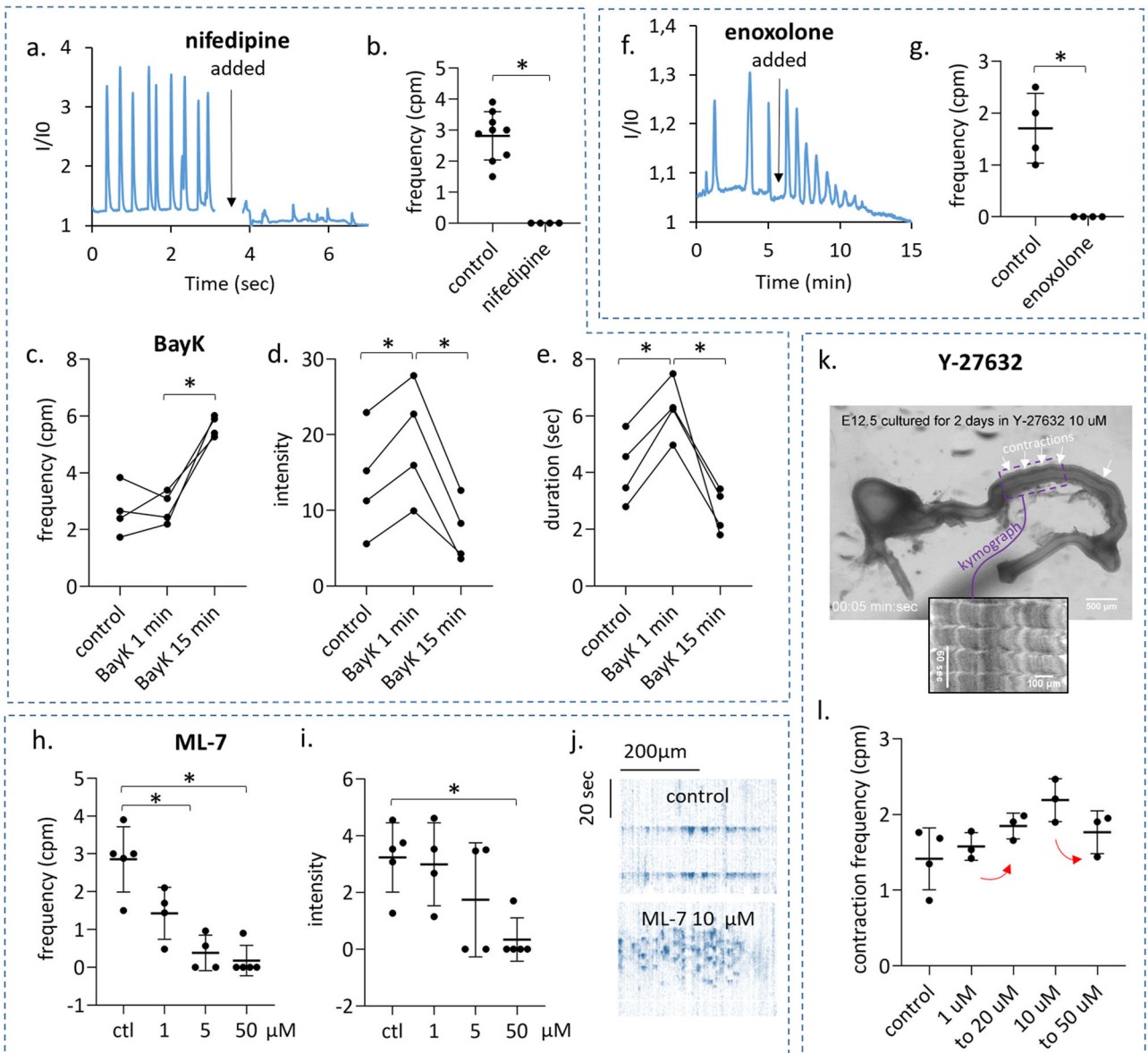

**Fig. 4 | Molecular determinants of calcium wave generation and propagation in the E12.5 murine midgut. a, b** The Ca$_V$1.2 antagonist nifedipine (10 µM) abrogated CWs ($n = 4$), while (**c–e**) the Ca$_V$1.2 agonist Bayk 8644 (2 µM) stimulated waves ($n = 4$), with contrasting effects at short (1 min after application) and longer (15 min after application) term (Video S4). **f, g** The gap-junction inhibitor enoxolone (33 µM) gave rise to frequent waves immediately after application, but CW activity eventually vanished after ~10 min ($n = 4$, Video S5). **h, i** The MLCK inhibitor ML-7 lowered both the frequency and intensity of CW. **j** At 10 and 50 µM, a desynchronization of calcium events across the gut was observed upon application (Video S6),

as seen in this kymograph ($y$: time, $x$: position along the long axis of gut tract). CW vanished after ~20 min at 50 µM in 4/5 samples. **k** Contractile waves were still present in E12.5 + 2 guts cultured in 10 µM Rho/ROCK inhibitor Y-27632. The inset is a kymograph showing the rhythmic waves. **l** Contraction frequency of E12.5 + 2 samples: controls ($n = 4$) and in Y-27632 1 µM ($n = 4$) and 10 µM ($n = 3$). Frequencies did not significantly change after increasing the concentration to 10 and 50 µM, respectively (red arrows). *$p < 0.05$, Mann–Whitney two-tailed test. Each dot represents a different sample.

where the LSM forms, indicating that LSM differentiation inhibition by nifedipine did not result from cell death/toxicity. We could further show that the apoptotic effect of the Ca$_V$1.2 blockers on the CSM and ENS also occurred in E12.5 + 2 guts (Fig. S1). This contrasts with the effect at E11.5 + 2, where ENS and CSM morphology did not differ between control and nifedipine-treated samples (Fig. 5a). These data suggest that Ca$_V$1.2 blockers become toxic once the CSM and ENS start specifically expressing Ca$_V$1.2 at stage E13.5 (Fig. 2), at the start of or during culture.

We asked whether the mesenchymal cells that are inhibited from becoming LSM by nicardipine could differentiate into another major cell fate of the mesenchyme, the interstitial cells of Cajal (ICCs), that specifically express the KIT receptor in the gut. Nifedipine-treated E14.5 + 2 samples

exhibited a distinct population of KIT-expressing cells located at the outer periphery of the gut and at the level of the myenteric plexus (Fig. 6h, i) that were not or only sparsely found in control (Fig. 6g, i).

### Is LSM differentiation driven by CSM contractions?

CSM contractions induce circumferential compression and also longitudinal tensile stress of similar magnitude[26] in the embryonic gut. CSM contractions are inhibited by nifedipine in E14.5 guts, raising the possibility that the inhibition of LSM differentiation by nifedipine at E14.5 + 2 could be due to the absence of the cyclic, longitudinal tensile mechanical stress associated with CSM contractions. To test this hypothesis, we cultured E14.5 guts for 2 days in nifedipine while applying cyclic longitudinal stretch, with an amplitude of

**Fig. 5 | Nifedipine inhibits Ca²⁺ waves and motility, but not smooth muscle differentiation in E11.5 + 2 cultures. a** Representative whole mount IHC for α-SMA and βIII-tubulin (Tuj1) of the jejunum of control (DMSO vehicle only) and nifedipine treated E11.5 guts after 2 days in culture. Both samples exhibited similar CSM differentiation and ENS structure. Aggregates of Tuj1 at the mesenteric border could be found in both control and nifedipine samples. **b** Midgut (black, *n* = 10 controls, *n* = 8 nifedipine) and colon (red, *n* = 7 controls, n = 5 nifedipine) length covered by CSM in control versus nifedipine E11.5 + 2 sample—no significant difference could be revealed. **c** Measuring motility at E11.5 + 2 revealed that all control samples (*n* = 10) had developed contractile waves in the midgut, while they were absent with nifedipine (*n* = 8). The drug was thus effective at suppressing CW but did not interfere with differentiation. *\**p* < 0.05, Mann–Whitney two-tailed test. Each dot represents a different sample.

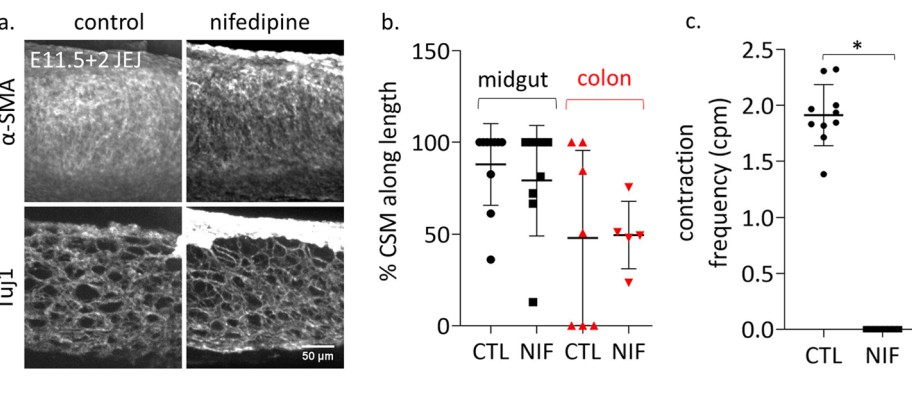

5–10%, at a frequency of 1 cpm, similar to the physiological contractions in the embryonic gut at these stages[27]. The stretch applied was sufficient to viscoelastically elongate the guts: the pins holding the lower part of the gut had to be repositioned twice a day to make sure the guts were still subject to tensile stress (cf. Fig. 7 inset and see the "Methods" section); at the end of the experiment, stretched guts were significantly longer than non-stretched ones. LSM coverage of cyclically stretched guts in nifedipine was extremely low (6 ± 4%, *n* = 4, Fig. 7) and not different from control guts kept in nifedipine without stretch (11 ± 5%, *n* = 7, Fig. 7): we could not recover LSM differentiation by cyclically stretching the guts in the presence of nifedipine.

## Discussion

We showed that CSM differentiation occurs in the mouse midgut in the interval E11.5–E12.5, and in the colon in the interval E12.5–E13.5. LSM differentiation starts in the duodenum and jejunum in the interval E14.5–E15.5. Both smooth muscle differentiation processes take place earlier in the more rostral regions of the intestine. Our chronology is in agreement with earlier, coarser (both temporally and spatially) investigations of α-SMA isoactin expression by in-situ hydridization[28]: this investigator found low to undetectable α-SMA at E11 in the midgut, but detectable levels at E13; the E11 colon was α-SMA negative at E11, and had moderate levels at E13; the LSM was found to differentiate 2–3 days after the CSM. Our results on LSM differentiation are consistent with a recent study in the jejunum[29]: these investigators found no LSM at E14, scattered LSM cells at E14.5 and E15, and a continuous LSM layer only from stage E16.

We found that spontaneous Ca²⁺ waves (CWs) were generated by the smooth muscle layer immediately after it differentiated, at E12.5 in the midgut and at E13.5 in the colon. This activity is purely myogenic: it is resistant to tetrodotoxin[21,30,31] and appears in cultured aneural chicken hindgut[21]. We could not detect any CWs before CSM differentiation, in E11.5 midgut or E12.5 colon. While this may seem to contrast with previous findings in a priori pre-differentiation E4 chicken[9], a closer look at the video provided by these investigators shows that there are distinct contractions accompanying the CW, suggesting that CSM had already differentiated in the sample they examined. We could also not detect CW in the midgut presumptive (E14.5) or already differentiated LSM (E14.5 + 2); at E14.5 + 2, activity in the LSM was present as the spiking of individual cells, and not as a synchronous, propagating wave. CSM CW were noticeably slower and less frequent in the colon than in the midgut. CWs at E12.5 do not induce contractions, or only very shallow ones in the more proximal regions of the midgut. The first detectable motility, i.e., inducing visible contractile waves propagating along the whole midgut, occurs one day after CSM differentiation, at E13.5 in the midgut and E14.5 in the colon[27,30]. The absence of

contraction at E12.5 could be due either to the relatively weak amplitude of the depolarization induced by the CW at this stage or to insufficient maturation of the contractile machinery. The first CWs at E12.5 propagate at a relatively high speed of ~350 μm/s; this speed then sharply decreases to ~30 μm/s at E14.5, and then increases again as development proceeds; in the adult mouse, myogenic contractile waves (slow waves) have a speed of ~1 cm/s[32]. The same non-trivial "U" shape of the evolution of contractile wave speed with development is also found in the chicken (deceleration between E6 and E9[21], acceleration after E12[33]). These changes in speed may reflect non-monotonous changes in the gap-junctional connectivity of cells during development, or in their threshold to Ca²⁺ release in the cytosol.

Consistent with previous findings on CW propagation in the early embryonic chicken intestinal smooth muscle[20,21], we found that CW in the mouse intestine relies critically on L-type Ca²⁺ channels and gap junctions. Ca$_V$1.2 channels were found to be expressed as from E11.5 in the outer gut layers, and its expression became restricted to the smooth muscle layer and even more prominently to enteric neurons as from E13.5. These findings are consistent with previous investigations reporting L-type Ca²⁺ channels in embryonic[34,35] and adult colonic[36] mouse enteric neurons. In later development, Ca$_V$1.2 channels play an important role in ENS spontaneous activity[35].

It is classically understood that contractility in skeletal and smooth muscle is triggered by Ca²⁺ entry through voltage-gated channels and the subsequent release of Ca²⁺ from intracellular stores. Two pathways are activated by this increase in intracellular Ca²⁺: on the one hand, calmodulin activates MLCK, and on the other hand, RhoA-Rho kinase inhibits MLCP. MLCK and inhibited MLCP jointly lead to phosphorylation of MLC20 that initiates actin-myosin stepping leading to contraction[23,24]. We expected that breaking this cascade at the level of MLCK with ML-7 would inhibit contractility, but not upstream Ca²⁺-related events. We find that ML-7 does, however, perturb CWs, by reducing their frequency at low concentrations and by disorganizing the physiological synchrony of the Ca²⁺ uprise across the mesenchyme at higher concentrations. These results show that the above-described cascade of events is not unidirectional, but that perturbations at any level feedback upstream, perturbing the whole Ca²⁺ handling chain within the cell. We further found that inhibition of Rho with Y-27632 did not affect contractility in the early-developing mouse gut, as had been assumed by other investigators[11]. The Rho pathway is known to be involved in adult gut smooth muscle contraction, particularly of its tonic component[24]; our results indicate that this pathway is not required for embryonic gut phasic contractility.

CWs were found to play an important role in drosophila wing[17], chicken feather bud[18] and mouse limb[19] development. Cytoplasmic Ca²⁺ ions entering through voltage-gated channels can act as an

**Fig. 6 | L-type Ca²⁺ channel blockers inhibit LSM differentiation, gut elongation, and promote differentiation of KIT-expressing cells at the gut periphery. a** Fraction length of E13.5 + 3 and E14.5 + 2 midgut covered by LSM for control and different concentrations of nicardipine (nic) or nifedipine (nif), in µM. **b** CSM contractile wave frequency at different concentrations of nicardipine: complete inhibition of motility was only achieved at 10 µM. **c** Correlation between the absence (red small dashed circle) or presence (blue large dashed oval) of motility and the elongation of E13.5 guts cultured for 3 days. **d**, **e** Tuj1, α-SMA and cleaved caspase-3 frozen section IHC of control (top) and nifedipine treated E14.5 + 2. Nifedipine abolished the differentiation of the LSM. It perturbed the structure of the ENS and of the already differentiated CSM and presented with more apoptosis in these regions than controls. Apoptosis was, however, not present in the presumptive LSM layer. **f** The fraction area of tissue with high apoptosis rates was significantly higher for nifedipine-treated E14.5 + 2 guts (n = 4) than in controls (n = 4). Each dot represents a different sample and is the average apoptotic area fraction of n = 3 slices examined. **g**, **h** DAPI, Tuj1, α-SMA, and KIT frozen section IHC of control and nifedipine treated E14.5 midgut. The LSM layer is absent in nifedipine-treated samples, and a distinct KIT+ cell population emerged at the outer gut periphery, in the region of the presumptive LSM. **i** c-KIT positive cells per mm² mesenchyme in control (n = 7) and 10 µM nifedipine treated (n = 8) E14.5 + 2 gut. Each dot represents a different sample and is the average of n = 3 slices examined. *p < 0.05, Mann–Whitney two-tailed test.

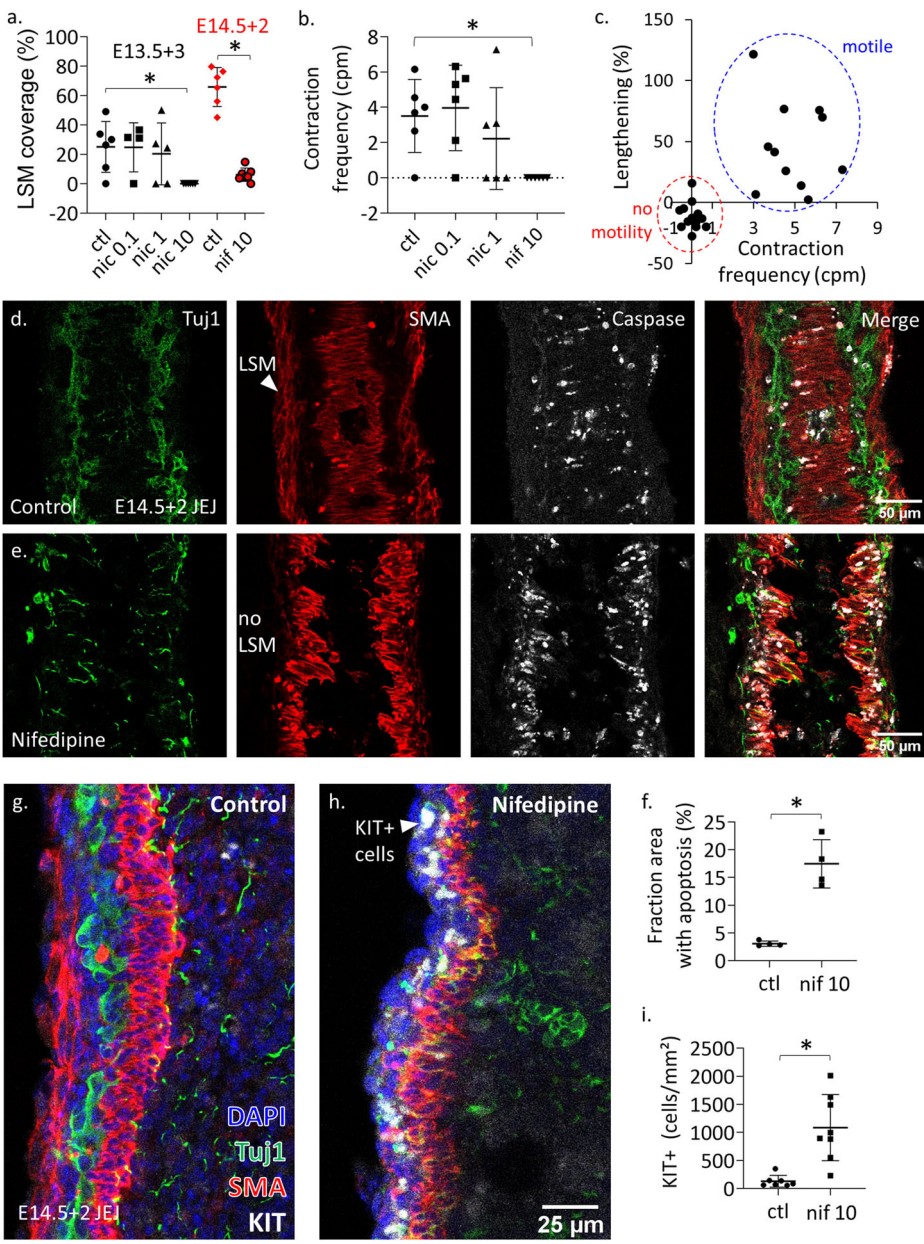

intracellular second messenger, activating several downstream signals, such as the calmodulin (CaM)/calcineurin and the PKC/MAPK pathways[19]. We found that early CSM CW waves are not required for CSM differentiation or development in the intestine, as their inhibition by nifedipine did not prevent CSM differentiation, nor alter the morphology of this layer.

At later stages, when CSM is already differentiated and contractile, we found that nifedipine inhibited LSM differentiation, and also halted elongation of the cultured intestines. These results extend to a mammalian model previous conclusions by our group[14,37,38] on the essential role of CSM contractility in gut lengthening: CSM contractions "squeeze" the gut cyclically[26], inducing longitudinal deformation and anisotropic growth of the organ. The YAP1 pathway has recently been shown to mediate the proliferative response of the mouse intestinal tissue in response to mechanical stress[11]. Of note, these investigators used Y-27632 with the intent of inhibiting contractions, and did not find that it perturbed elongation: we now understand that Y-27632 does not actually affect CSM contractility in the embryonic mouse, which is why it also did not affect growth in their experiment[11].

The mechanism leading to the inhibition of LSM differentiation by nifedipine remains to be elucidated. It could be caused by a direct biochemical effect mediated by Ca_V1.2 in differentiating mesenchymal cells, breaking a morphogen cascade, possibly involving PDGFR[13]. But it could also be mechanically induced, because the CSM contractions and the associated cyclic mechanical stress they generate are inhibited by nifedipine. In chicken, 10 µM nifedipine treatment of E9 + 3 chicken guts did not abrogate differentiation but led to misorientation of the outer (LSM) muscle layer, which oriented circumferentially instead of longitudinally[9]. This misorientation could be rescued by applying cyclic longitudinal mechanical stress. The experiments we performed in mice show several differences to these findings in chicken: (1) α-SMA staining was mostly absent in the presumptive LSM region in the presence of nifedipine, i.e. differentiation was affected, (2) we did not observe misorientation of the LSM in mouse guts treated with nifedipine in the few regions where LSM was still present (~10% of gut length), (3) we could not rescue LSM differentiation by cyclic longitudinal stretching in the presence of nifedipine. These discrepancies could be plainly related to differences in the mechanisms of myogenesis between chicken and mice. We also note that Huycke et al. did not report the extent of gut length covered by the misoriented LSM fibers in nifedipine

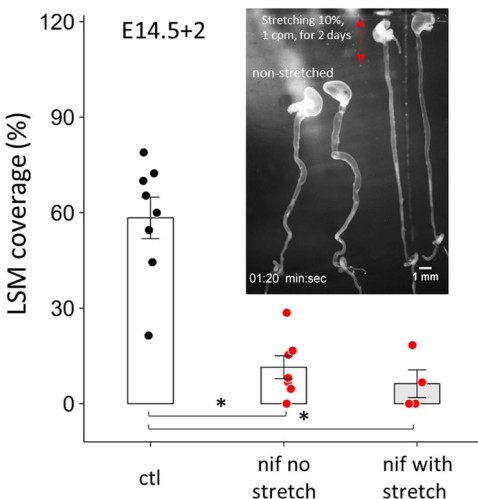

**Fig. 7 | Cyclic longitudinal stretching of E14.5 + 2 guts in nifedipine did not rescue LSM layer formation.** Fraction gut length covered by LSM in controls (without stretch), nifedipine 10 μM without stretch and nifedipine 10 μM with cyclic stretch. Inset shows a picture of the experiment where the two guts on the right are pinned at the level of the cecum (lower right corner), and a cantilever is inserted in the stomach. The cantilever moved up and down at 1 cpm to exert ~10% stretch during 2 days in culture. Two non-stretched guts in nifedipine are seen on the right. *$p < 0.05$, Mann–Whitney two-tailed test. Each dot represents a different sample.

conditions, i.e., perhaps some inhibition of differentiation took place in their experiments as well. Finally, our mechanical stimulation experiment does not completely rule out the possibility that LSM differentiation is induced by CSM contractions. The cyclic stretch we applied was sufficient to significantly elongate the guts (compared to non-stretched guts in the same bath), but we cannot exclude that the amplitude (5–10% at 1 cpm, with re-stretching of the preparation twice per day) may have been insufficient to continuously stimulate LSM differentiation.

Besides contractile smooth muscle, mesenchymal cells in the gut can also differentiate into pacemaker interstitial cells of Cajal (ICCs), fibroblasts, or blood vessels of the presumptive mucosa. Many gastrointestinal diseases, like chronic intestinal pseudo-obstruction[39] or diabetes[40], are characterized by a lack of intestinal rhythmicity and a depletion of the ICC pool[41–43]. It is, therefore, of clinical importance to understand how a normal SM–ICC balance can be restored in these pathologies. We found that nifedipine led to a 10× fold increase in c-KIT-expressing ICC cells in the peripheral intestinal mesenchyme of E14.5 + 2 mouse embryos. It is well-known that the electric pacemaking activity of ICCs is not dependent on L-type $Ca^{2+}$ channels[44,45]. Here we show that L-type $Ca^{2+}$ channel activity is not required for ICC differentiation; it is likely that the cells, being inhibited from becoming smooth muscle, turn to the other major mesenchymal phenotype, ICCs. This effect is similar to that of PDGF receptor blocking, which inhibited LSM differentiation and instead triggered ICC differentiation at the outer gut periphery[13]. Recently, we showed that PDGFRA expression was associated to CIPO SMCs[46] due to a phenotypic switch of the CIPO-SMC towards an undifferentiated cell state. The capacity of gastrointestinal, but also vascular and urogenital SMC, to plastically alter their phenotype between a differentiated/contractile and an immature/proliferative phenotype under exogenous or endogenous stimulation can lead to important functional alterations[2,47], and are an important subject of future investigation.

## Methods
### Ethics
All experiments were carried out according to the guidelines of CNRS and INSERM animal welfare committees, and conform to their principles and regulations. We have complied with all relevant ethical regulations for animal use. Mice were hosted at the Institut Jacques Monod animal

husbandry, and had access to housing and food and water ad libitum. Pregnant mice were killed by cervical dislocation to retrieve embryos age E11.5–E14.5. The embryos were separated and immediately beheaded with surgical scissors. The sex of the embryos was not assessed. The methods used to kill the mice/embryos conform to the guidelines of CNRS and INSERM animal welfare committees. Killing of mice for retrieval of embryos and/or their gastrointestinal tract is a terminal procedure for which neither CNRS nor INSERM assign ethics approval codes and hence none are given here.

### Mouse gut samples
The Cre reporter mice C57BL/6N-Gt(ROSA)26Sor[tm1](CAG-GCaMP6f) Khakh/J mice, referred to as Gcamp6fl/fl, were obtained from The Jackson Laboratory (Stock No. 029626). A transgenic mouse line in which the transgene is under the control of the 3-kb fragment of the human tissue plasminogen activator (Ht-PA) promoter Tg(PLATcre) 116Sdu16, was referred as Ht-PA::Cre. GCamp6fl/fl males were crossed with Ht-PA::Cre females to generate embryos carrying the $Ca^{2+}$ fluorescent reporter in neural crest cells and their derivatives. In ~25% of the embryos, the GCamp6f signal was located in the mesenchyme, but not in the NCCs. These mesenchyme-specific GCamp6f reporter mice were used for $Ca^{2+}$ imaging in this study; neural-crest specific GCamp6f expressing guts were used for culture experiments. We could clearly distinguish the two types of samples as neural crest specific guts (Video S8) generated a signal that outlined the developing enteric nervous system, and was characterized by local, uncoordinated sparks of $Ca^{2+}$ activity restricted to the ENS, and not mesenchyme-spanning waves (Video S1–S6). For some experiments where the GCamp6f reporter was not required, we also used wild-type C57Bl/6N mice. Animals were dissected at stages E11.5–E15.5, embryos removed, and the gut of each embryo was dissected in PBS from stomach to colon.

### Calcium imaging and analysis
Each gut was immobilized in 1 mL of 1% low-melting point agarose (Sigma A4018) in a 35 mm diameter Petri dish (Greiner), and covered with 4 mL of DMEM:F12 Glutamax (Gibco 31331-028) with 1% penicillin–streptomycin and 15 mM HEPES. Guts were imaged on a thermostated support at $36.5 \pm 1$ °C on a Zeiss LSM 780 upright confocal microscope with a ×20 objective at illumination wavelength 488 nm, and fluorescence was collected in the range 492–630 nm at a 2 Hz acquisition rate. Some videos were acquired on an inverted spinning disk confocal microscope (Olympus). Drugs were added from stock solutions directly in the Petri dish under the microscope and homogenized by up-down movements with a 1 mL pipette. The effect of each drug concentration was assessed for at least 3 min. Drugs included: nifedipine (Sigma N7634, stock 10 mM in DMSO), nicardipine (Sigma N7510, stock 10 mM in DMSO), (S)-(-)-Bay K8644 (Sigma B133, stock 10 mM in DMSO), ML-7 (Tocris 4310, stock 50 mM in DMSO), Y-27632 (Tocris 1254, stock 10 mM in DMSO), enoxolone (Abcam 142579, stock 33 mM in DMSO). Analysis of CWs were performed by drawing a region-of-interest (ROI) and plotting the pixel intensity against time; we computed the frequency as the number of intensity peaks per unit of time extracted from 3 to 5 min videos, and the average intensity ratio as the average $I_{peak}/I_{baseline}$ for all peaks, using the Matlab "findpeaks" function. Amplitude measurements were compared for the same region of the same sample, keeping the same ROI size, but in different pharmacological conditions. They were performed at stage E12.5 where contractions are either absent or very weak and do not perturb the measured fluorescence signal.

### Organotypic culture and morphometrics
E11.5, E12.5, and E13.5 mouse guts were cultured free-floating in DMEM:F12 Glutamax (Gibco) with 1% penicillin–streptomycin in individual 35 mm diameter Greiner culture dishes, for 2–3 days in a humidified incubator at 37 °C, in a 5% $CO_2$–95% air atmosphere. For bulkier E14.5 guts the atmosphere was 5% $CO_2$–95% $O_2$. To assess elongation, the length of the guts was measured prior to and post-culture from binocular images (MZ series, Leica) with the ImageJ "segmented line" function.

## Cyclic stretch culture

Cyclic stretch culture was performed in a 40 mL optically transparent trough covered with a vertical sheet of PDMS (~5 mm thick). Control E14.5 (non-stretched) guts were pinned at the stomach and cecum to the PDMS sheet, without applying stretch. The stomach of the stretched E14.5 guts was first attached to a thin (~0.5 mm) horizontal glass fiber obtained by heat-pulling a Pasteur pipette. The cecum was then pinned to the PDMS in such a way that the gut was fully elongated (but not stretched) in the vertical direction. The Pasteur pipette was fastened to a micro-stepper motor (Newport M-UTM linear stage with ESP300 controller), controlled via an RS232 interface and a Matlab program. Each stretch cycle corresponded to a 2–3 mm upward movement of the glass fiber (to which the stomachs are attached) at a speed of 2 mm/s, a 2 s pause in the stretched state, and a subsequent downward movement of 2–3 mm at 2 mm/s back to the initial state. The strain computed from $(l_{stretched} - l_{relaxed})/l_{relaxed}$ was in the range 5–10%. The stretch was applied at 1 cpm, corresponding approximately to the frequency of spontaneous CSM contractions in the mouse gut[27]. Because this cyclic stretch gradually increased the length of the intestine, we repositioned the upper fiber 2 times per day to correct for this. Guts were harvested after 48 h and processed for IHC as described below.

## Monitoring of gut motility

Spontaneous motility of mice gut post-culture was monitored by placing them in 1 mL of medium in individual dishes at 37 °C and time-lapse imaging with a binocular (MZ series, Leica) and camera (Stingray) at a 1 Hz frequency for 2 min. Space–time diagrams of motility[21] in different regions were extracted with the imageJ "Reslice" function.

## Immunohistochemistry

For whole mount IHC, post-culture samples were fixed for 1 h in 4% PFA in PBS, washed 3 times, then blocked and permeated in 1% BSA and 0.1% triton in PBS overnight, immersed in 1:500 anti-α-SMA Cy3 conjugated antibody (Sigma C6198) and 1:500 anti-βIII-tubulin FITC conjugated antibody (ab224978) for 24 h, washed, immobilized in 1% agarose gel (Sigma A4018) in PBS and imaged (Zeiss LSM 780). For assessment of the percent fraction of gut length covered by CSM or LSM we imaged the gut along the midgut length from proximal duodenum to the ileo-cecal junction and we computed the ratio of length where these muscular layers were present over the total midgut length.

For frozen section IHC, samples were dehydrated for 24 h in 30% sucrose solution in water, and then embedded in OCT. Frozen sections were performed on a Leica cryotome at 20 μm thickness. Sections were blocked and permeated in 1% BSA and 0.1% triton in PBS overnight. They were then incubated for a day in primary antibody, washed, and incubated for another day with the secondary antibodies and conjugated antibodies. Primary antibodies used were rabbit anti-Ca$_V$1.2 (Alomone Labs ACC003, dilution 1:100), rabbit anti-cleaved caspase-3 (Cell Signaling Technology, 9664, dilution 1:500), rabbit PH3 (anti-histone H3 phospho S10) (Abcam 14955, dilution 1:200). We used anti-rabbit AF647 secondary antibody (Thermofisher, dilution 1:400) to reveal the non-conjugated primaries. Conjugated antibodies included the α-SMA Cy3 and βIII-tubulin FITC mentioned above, and c-KIT-AF647 conjugated antibody (1:100, Santa Cruz Biochemicals sc-393910). We quantified the surface density of KIT positive cells by drawing a region-of-interest around the gut tunica muscularis (LSM + ENS + CSM), visually counting the number of KIT+ cells in this ROI, and dividing the count by the area of the ROI; the same procedure was also adopted for proliferation assessment with PH3. Since apoptotic cells could be very numerous, especially in nifedipine-treated samples, we quantified rather the fraction area of tissue where dense cleaved caspase-3 staining was present.

## Statistics and reproducibility

All sample numbers indicated in this report correspond to different embryos (guts, biological replicates). All experiments involving culture were technically replicated at least twice, i.e. they were performed on two different days (different litters) with fresh samples following the same procedure each time. A minimum of $n = 4$ samples constitutes each group presented in this report, except in Fig. 2 where only $n = 2$ embryos were used per stage. Statistical analysis was performed with the two-tailed Mann–Whitney test and was considered significant at $p < 0.05$.

## Reporting summary

Further information on research design is available in the Nature Portfolio Reporting Summary linked to this article.

## Data availability

All data generated or analyzed during this study are included in the manuscript and supporting files. The source data underlying the graphs and charts can be found in Supplementary Data 1. Other data are available from the corresponding author upon request.

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

## Acknowledgements

This research was funded by the Agence Nationale de la Recherche ANR GASTROMOVE-ANR-19-CE30-0016-01, by the Université de Paris IDEX Emergence en Recherche CHEVA19RDX-MEUP1, by the CNRS PEPS INSIS "COXHAM" grant, by the Labex "Who AM I ?" ANR-11-LABX-0071, and by the Imaging platform BioEmergences-IBiSA, ANR-10-INBS-04 and ANR-11-EQPX-0029. We are grateful to Sylvie Dufour for providing us with the Ht-PA::Cre mouse line.

## Author contributions

N.R.C. led the project, obtained funding, performed experiments, analyzed data, synthesized data; P.d.S.B. and S.F. performed experiments and analyzed data; L.Z., A.G., R.A.A., A.E.L., L.P., I.L.P., N.R., I.A. performed experiments; N.R.C. wrote the paper; N.R.C., P.d.S.B., and S.F. revised the paper.

## Competing interests

The authors declare no competing interests.
