## [Transparent Peer Review file · Communications Biology]

Calcium wave dynamics in the embryonic mouse gut mesenchyme: impact on smooth muscle differentiation

Corresponding Author: Dr Nicolas Chevalier

Version 0:

Reviewer comments:

Reviewer #1

(Remarks to the Author)

Smooth muscle cells are the contractile machinery of gut motility and differentiate from mesenchyme. Nicolas et al. employed GCamp6f-expressing mouse embryos to investigate the effects of calcium signals on the development of the inner circular smooth muscle (CSM) layer and outer longitudinal smooth muscle (LSM) layer. The authors found that pharmacologically inhibiting calcium waves halted the differentiation of CSM and LSM, as well as intestinal growth. Furthermore, they observed that mechanical stimulation did not rescue LSM differentiation when the calcium waves were inhibited by nifedipine. This led them to conclude that the effects of calcium waves on smooth muscle development are not mediated by contractile force. Finally, the author investigated the effects of the inhibition of calcium waves on the cell composition change in chicken embryos and found besides the retard in smooth muscle differentiation, the number of interstitial cells of Cajal was increased.

The manuscript's topic is interesting and important in the context of gut development. However, there are several major concerns in the present manuscript.

1. The Methods section stated that "In ~25% of embryos, the GCamp6f signal was localized in the mesenchyme, but not in the NCCs. These mesenchyme-specific GCamp6f reporter mice were used for calcium imaging in this study; neural crest-specific GCamp6f expressing intestines were used for culture experiments." The rationale for this experimental approach is unclear and would benefit from further explanation.
2. While the analysis of calcium waves by drawing a region of interest (ROI) over the region (either the putative CSM or LSM) is appropriate for quantifying frequency, it may present challenges for amplitude comparisons. Variations in ROI sizes could result in different amplitude values for the same underlying calcium signals. Additionally, it should be noted that each calcium wave is associated with a contraction. Therefore, it is advisable to reanalyze calcium amplitudes that track CSM or LSM over time.
3. The study relies heavily on pharmacological manipulations involving L-type channel inhibitors (nifedipine and nicardipine) and the agonist BayK, all of which inhibit calcium waves and contractions. Although the authors mentioned a time-dependent change in calcium signals caused by BayK, these pharmacological experiments may weaken the authors' claim that L-type calcium channels are involved in calcium waves, contractions, and embryonic gut growth.
4. In previous work, the authors demonstrated that CSM contractility is essential for gut elongation and morphogenesis, as discussed in paragraph 3 of the Discussion. In the present study, the authors showed that the effect of calcium waves on intestinal differentiation is not mediated by mechanobiological pathways related to smooth muscle contractions. However, it seems that an additional control group (ctl+stretch) should be included before ruling out the role of stretch or contraction in intestinal differentiation.
5. The choice of experimental models appears inconsistent, with mice used for the initial investigation of calcium waves and mechanical stimulation on intestinal smooth muscle differentiation (Figures 1-4), followed by a switch to chicken embryos (Figures 5a-e), and a return to mice (Figures 5f-h). Additionally, the authors should specify the method for quantifying immunostaining in Figures 5f and 5g. The current images in these panels are blurry, making it challenging to ascertain whether the dots indicated by arrows represent individual cells or cell clusters.
6. In Figures 5a to 5d, the part in parentheses "(relative to control)" in the y-axis title should be revised to "(relative to GAPDH)" or "(relative to RPLPO)" for clarity and accuracy.

Reviewer #2

(Remarks to the Author)

In this manuscript Chevalier et al., analyzed in mouse and chicken embryo the effect of calcium waves to smooth muscle differentiation. The authors applied multiple organ cultures combined with embryonic gut motility analysis, calcium imaging for smooth muscle and enteric neurons, and immunocytochemistry to characterize the development of smooth muscle and motility before birth/hatching. Greater understanding of the intestinal motility and emphasizing the complex molecular and mechanobiological pathways preceding the visceral smooth muscle differentiation are a great value to both congenital gastrointestinal diseases and muscle biology field. The presented findings provide mechanistic insight into how calcium waves are required for the smooth muscle differentiation in the embryonic gut.

Comments and recommendations:

- 1) last paragraph of the Introduction section is highly similar to the Abstract.
- 2) Introduction, first paragraph: muscularis mucosae "is located between lamina propria and submucosa layers" instead of "under the epithelium"
- 3) Introduction, fourth paragraph: *Drosophila melanogaster* instead *D.Melanogaster*
- 4) Results (first paragraph + Fig1). According to previous papers smooth muscle differentiation starts around E11.5 in the mouse intestine (Sharbati, 1982, *J Anat*; MCHugh, 1995). This contradicts the authors' statement about smooth muscle development. More specifically, according to McHugh, 1995 (paper also cited in this manuscript), alpha-smooth muscle isoactin gene expression is first evident in the E11.5 hindgut. In contrast of cranio-caudal differentiation pattern of the embryo the intestinal smooth muscle differentiation initiates from both caudal and cranial end of the gastrointestinal tract. This should be discussed.
- 5) To clarify the controversy, I would recommend for the authors to include to Fig 1 additional images from 12.5 dpc midgut and 13.5 dpc hindgut cross-sections immunostained with alpha-smooth muscle actin antibody.
- 6) What is the source of calcium waves, it is produced by mesenchymal cells or enteric neurons? The E12.5 (dpc) jejunum analyzed in Video 1 should already contain enteric neural crest derived cells. Do you see calcium waves in the aganglionic segment of the gut? To generate experimental aganglionosis in mice you can culture the E10.5 postumbilical midgut or E11.5 hindgut for 2 days and analyze the presence of calcium waves.
- 7) ... "the nifedipine treated gut samples were also affected, as Tuj1+ cells were scarce, no LSM differentiation, motility was impeded.... How the intestinal proliferation and apoptosis rate changes after nifedipine/nicardipine treatments? To increase the quality of their work the authors should consider about including anti-casp3/Tuj1 and anti-casp3/alpha-SMA double immunos to Fig 2.
- 8) How the ENS patterning and differentiation is affected by nifedipine/nicardipine treatments?
- 9) Fig S2f is not explained in Results and Legends sections. Which segment of the chicken gut is shown on Fig S2a-c?
- 10) Fig S2f labeling: % "dead" cells.... you mean death? How was this analyzed? Describe the method.
- 11) Clone name for the Tuj should be Tuj1.
- 12) Please includes what is the molecule recognized by Tuj1 mAb.
- 13) It is not clear why the Rho/Rock inhibition had to be done?
- 14) Fig.4b size (labeling, column size) is at least twice bigger than Fig. 4a. You should resize the image and diagram.
- 15) Page 8. Chicken experiment: which part of the intestine was cultured for 3 days, and analyzed for ACTA2 and ACTG2 expression?
- 16) "As expected, at E5, the intestinal mesenchymal is not yet differentiated." ---According to Graham et al 2017, *J Anat*, E5 chicken midgut is alpha-SMA negative, hindgut is alpha-SMA immunoreactive. This controversy should be mentioned/clarified and discussed.
- 17) Fig.5 f and g. Please show sperate channels for DAPI and SMA, KIT antibodies (blue, green, red) because the c-Kit immunoreactive cells are not seen in Fig. 5g. Also include insets or separate figures from high power pictures to prove the presence of c-KIT ICC cells on this section. The background is very high maybe caused by the anti-mouse c-KIT antibody detected with anti-mouse secondary on mouse sections. I would recommend this antibody: rat anti-c-Kit (ACK2, 1:800, eBioscience, USA) used by <https://link.springer.com/article/10.1007/s00441-019-03120-9#Sec2> to specifically stain the mouse c-KIT+ ICCs

Reviewer #3

(Remarks to the Author)

Summary: This study uses GCaMP6f mice to show that calcium waves propagate in mouse intestinal mesenchyme prior to α SMA expression in the smooth muscle layer, and L-type calcium channel may be the main calcium channel responsible. The authors tried to use ML-7 and Y-27632 to delineate the involvement of mechanical force (stretching, myosin contraction) in this calcium propagation, but due to the flaws in some experimental design, the conclusion is not fully supported. However, there are some new findings, and calcium signaling in smooth muscle differentiation is an important topic. So the work is valuable for this journal. Please address the following to improve.

Comments:

1. One of the strengths of this paper is the live imaging of calcium propagation as demonstrated in Fig 1. However, the authors did very little to explore and investigate the effects of perturbing calcium signaling to CSM or LSM differentiation. For example, how do different doses of nif affect smooth muscle pattern formation? How does Nif and BayK treatment affect smooth muscle formation differently? How does Nif/BayK treatment affect the orientation of smooth muscle cells?
2. What is the L-type calcium channels expressing in the LIX1 progenitor mesenchymal cells? Will the expression change during the differentiation process to CSMs, LSMs, and ICCs?
3. Is the pre-CSM and pre-LSM calcium wave also mediated by gap junction as the calcium waves inducing early contractile events? It could be tested by inhibitor of gap junctions. And if so, which connexins are expressed and functioning here?
4. Fig 3. One of the most effective doses of ML-7 on frequency is 5 μ M, but its effect is not included in the video. Why is 10 μ M less effective than 5 μ M?
5. In figure 4b, to better examine the relationship between mechano-stretch and LSM differentiation, one more group could be added as "stretch and no inhibitor" – to check if stretch itself could influence LSM differentiation - it may go through other mechano-sensation channel but not L-type calcium channels.
6. What is the signaling pathway calcium take to promote LIX1+ cells differentiating to KIT+ ICCs but not smooth muscle lineages? Could the author try inhibitors of potential downstream target of calcium signaling pathway (e.g., calmodulin inhibitors)?
7. Fig 4b. Cyclic stretch: a stretch only group is needed to show the experimental setup is done properly by showing an effect on LSM coverage.
8. In figure 5, the name of the figure is better changed to a statement instead of a question. Also, panel f and g seems to be under different scale - a scale bar for panel g should be added.
9. Relying on a couple markers to claim a change in cell fate is insufficient in the omics era. Some basic bulk-RNaseq should be performed to support and show the overall change in cell fates when calcium is perturbed.
10. Ctl video is missing for Y-27632 treated sample

Version 1:

Reviewer comments:

Reviewer #1

(Remarks to the Author)

The revised manuscript is much improved. The authors now perform all experiments in mice, which makes the data more coherent and the results more interesting. However, several issues still need to be addressed:

1. In several places, the authors claim that Ca^{2+} waves are mediated by Cav1.2. Without genetic manipulation or an isoform-specific L-type Ca^{2+} channel inhibitor, this conclusion seems premature. Nifedipine and Bay K8644 do not act exclusively on Cav1.2. Can the authors rule out the involvement of Cav1.3 or Cav1.1 in these Ca^{2+} waves?
2. Figures 1 and 2: The authors should use more objective imaging analysis. Data are presented descriptively. How confident are the authors that there is no α -SMA, and therefore no smooth muscle fibers, at E11.5? The images show robust signals in all three parts of the gut at E11.5. What are the negative controls for these immunostains?
3. Figure 5: In the nifedipine group, both α -SMA and Tuj1 were significantly upregulated. Are these signals artifacts? If they are real, the authors should comment on their relevance.
4. Figure 7: Since the guts were stretched longitudinally, these experiments do not provide convincing evidence to answer the question "Is LSM differentiation driven by CSM contractions?"

Reviewer #2

(Remarks to the Author)

The authors have done a superb and conscientious job of meeting all of the criticisms of their initial submission. The paper is much improved and needs no further revision.

Reviewer #3

(Remarks to the Author)

Authors have done a lot work to revise and the manuscript is significantly improved. They have addressed most of my concerns. Thank you.

General response to all reviewers

We thank the referees for their reports and constructive criticism of our work. We have performed many new experiments following this first round of review, which have led us to drastically change the interpretation of our results and ensuing conclusions. We are summarizing below the most important changes:

1°) Following Ref. 2 comments on the chronology of smooth muscle differentiation, we have performed whole-mount and frozen slice IHC of α -SMA at stages E11.5-E14.5. It appears that **SMA differentiation takes place one day earlier than we initially thought, at E12.5 in the midgut, and E13.5 in the colon.** Our initial assessment was based on a series of whole-mount IHCs performed several years ago, and although we struggle to pinpoint exactly where our error stems from (older microscope ? unsuccessful whole-mount IHC ? uncertainty in the age of the embryos ?), it appears we overlooked the presence of fibers in the MG at E12.5 at the time. We were moreover misled by the fact that E12.5 guts do not display any contractions / motility, which we thought was a sign that SMA was not yet present ; we now understand that in the mouse (unlike in the chicken), motility arises one day after differentiation of the SMA ring.

2°) This means that the Ca^{2+} waves at E12.5 that we initially interpreted as mesenchymal, pre-differentiation waves, are actually the first Ca^{2+} activity of the smooth muscle immediately after differentiation. **We do not detect any activity before differentiation**, be it in the E11.5 midgut, or in the E12.5 colon.

3°) We nonetheless investigated whether this early Ca^{2+} activity is important for differentiation, repeating the nifedipine experiment starting at stage E11.5, for 2 days in culture. The results do not leave a doubt: nifedipine completely abrogates Ca^{2+} waves (and motility), but the circular SMA layer (CSM) does differentiate, indicating that these **early Ca^{2+} wave are dispensable for CSM formation** (Fig.5).

4°) How then do we explain the inhibiting effect on CSM we previously reported at E12.5+2? The cleaved-caspase3-IHC experiments suggested by Ref.2 revealed that nifedipine massively induced **apoptosis in post-differentiation SMA**. The new images we present clearly show that the regions where CSM is not present anymore in E12.5+2 guts display apoptosis (Fig.S1). This apoptosis is also present in E14.5+2 CSM (although it does not cause the layer to disappear completely as at E12.5+2), and on E12.5+2 and E14.5+2 enteric neurons. Interestingly however, we do not find similar apoptosis in E11.5+2 SMA or enteric neurons. Following Ref.3 comments, we now provide staged IHCs of L-type Ca^{2+} channels and find that they are diffusely expressed at E11.5-12.5, and conspicuously co-localize with the CSM and the enteric neurons as from E13.5. This suggests that the apoptotic effect of nifedipine only takes place when these cells start expressing L-type Ca^{2+} channels (at the beginning or during culture).

5°) Could the inhibition of LSM formation at E14.5+2 also be due to apoptosis ? The response is no: cells at the outer periphery of the intestine are not apoptotic. We moreover could confirm the distinct expression of KIT (instead of LSM) induced by nifedipine in 5 additional samples (2 controls, 3 nifedipine). We present this now more clearly in Fig.6.

6°) How do we fit our “chicken results” in the picture ? We had found that nicardipine treatment at E5+3 led to a decrease of CSM coverage and of ACTG2 mRNA levels. As suggested by Ref.2 on the presence of α -SMA in the hindgut [1,2] at E5, it seems likely that the decreased ACTG2 mRNA levels we measured were due to the same apoptotic effect of nicardipine on the already differentiated SMA. For the sake of clarity, we have therefore found it preferable to not include our results on chicken in this revised version.

We would like to outline the key points and novelties of our new manuscript:

- We provide a **precise and consistent chronology of α -SMA differentiation** at stages E11.5-E15.5 by whole-mount (Fig.1) and frozen slice (Fig.2) IHC, both methods being consistent.
- As suggested by Ref. 3, we now present **staged IHCs of L-type calcium channels (CaV1.2)**, demonstrating distinct expression by the CSM layer in the interval E12.5-E13.5, and by enteric neurons as from E13.5 (Fig.2). It is particularly interesting that enteric neurons express CaV1.2 (even more conspicuously than muscle), and we have recently confirmed this and the functional importance of this channel at later stages [3].
- We are providing the **first videos of Ca^{2+} waves propagation in the differentiating midgut and colonic mesenchyme** at stages E11.5 (no waves), E12.5, E14.5 and E14.5+2, along with their characteristics – we strived to make this material as illustrative as possible videos S1 and S2 and Fig.3.
- We provide fundamental information on the **molecular underpinnings of the Ca^{2+} waves** (Fig.4):
 - they are CaV1.2 (inhibited nifedipine, excited by BayK) and gap-junction (inhibited by enoxolone) dependent
 - they are disorganized and eventually inhibited by the MLCK-inhibitor ML-7, showing that inhibiting contractility at the myosin level feeds back on the whole upstream Ca^{2+} handling chain; we could have expected waves to propagate without ensuing contractions in ML-7, but this is not the case
 - they do not depend on the Rho/ROCK pathway at these early stages, as they are unaffected by Y-27632.
- We demonstrate that Ca^{2+} waves are only present after CSM differentiation, and are dispensable for CSM differentiation (Fig.5).
- We demonstrate that **nicardipine abrogates LSM differentiation, instead triggering the emergence of KIT expressing ICC cells** (Fig.6).

We thus believe the report adds a wealth of new information on intestinal smooth muscle development. The referee comments have been instrumental in the proper interpretation of our findings. We'd like to thank you for your feedback, and hope you will find the present version satisfactory.

Reviewer #1 (Remarks to the Author):

Smooth muscle cells are the contractile machinery of gut motility and differentiate from mesenchyme. Nicolas et al. employed GCamp6f-expressing mouse embryos to investigate the effects of calcium signals on the development of the inner circular smooth muscle (CSM) layer and outer longitudinal smooth muscle (LSM) layer. The authors found that pharmacologically inhibiting calcium waves halted the differentiation of CSM and LSM, as well as intestinal growth. Furthermore, they observed that mechanical stimulation did not rescue LSM differentiation when the calcium waves were inhibited by nifedipine. This led them to conclude that the effects of calcium waves on smooth muscle development are not mediated by contractile force. Finally, the author investigated the effects of the inhibition of calcium waves on the cell composition change in chicken embryos and found besides the retard in smooth

muscle differentiation, the number of interstitial cells of Cajal was increased.

The manuscript's topic is interesting and important in the context of gut development. However, there are several major concerns in the present manuscript.

1. The Methods section stated that "In ~25% of embryos, the GCamp6f signal was localized in the mesenchyme, but not in the NCCs. These mesenchyme-specific GCamp6f reporter mice were used for calcium imaging in this study; neural crest-specific GCamp6f expressing intestines were used for culture experiments." The rationale for this experimental approach is unclear and would benefit from further explanation.

Our main current project pertains to calcium signals in enteric neurons, which derive from the neural crest, which is why we designed this mouse-line. As we were experimenting, we noticed that ~25% of the embryos actually expressed GCaMP in the mesenchyme, as can be clearly seen from the data and supplementary videos we present. We do not currently have an explanation of why this happens. As proof of the fact that we could clearly distinguish these two types of samples we are providing now a Video S8 of the GCaMP expressed by the neural crest, which show a clearly different morphology and behavior (local sparks instead of waves). We modified the manuscript as such:

"We could clearly distinguish the two types of samples as neural crest specific guts (Video S8) generated a signal that outlined the developing enteric nervous system, and was characterized by local, uncoordinated sparks of Ca²⁺ activity restricted to the ENS, and not mesenchyme-spanning waves (Video S1-S6)."

2. While the analysis of calcium waves by drawing a region of interest (ROI) over the region (either the putative CSM or LSM) is appropriate for quantifying frequency, it may present challenges for amplitude comparisons. Variations in ROI sizes could result in different amplitude values for the same underlying calcium signals. Additionally, it should be noted that each calcium wave is associated with a contraction. Therefore, it is advisable to reanalyze calcium amplitudes that track CSM or LSM over time.

We only compared amplitudes for one given sample in the same region, keeping the same ROI size, but in different pharmacological conditions (Fig.4). E12.5 guts didn't contract, or only very weakly so, and thus did not affect the resulting intensity measurement. We did not perform amplitude measurements at later stages (E14.5, E14.5+2) where contractions cause movements of the sample out of the confocal plane that would clearly affect the measurements. We clarified this in the Materials and Methods:

"Amplitude measurements were compared for the same region of the same sample, keeping the same ROI size, but in different pharmacological conditions. They were performed at stage E12.5 where contractions are either absent or very weak, and do not perturb the measured fluorescence signal."

3. The study relies heavily on pharmacological manipulations involving L-type channel inhibitors (nifedipine and nicardipine) and the agonist BayK, all of which inhibit calcium waves and contractions. Although the authors mentioned a time-dependent change in calcium signals caused by BayK, these pharmacological experiments may weaken the authors' claim that L-type calcium channels are involved in calcium waves, contractions, and embryonic gut growth.

Our conclusions have indeed substantially changed following the different feedbacks of the reviewers and the additional experiments we have performed.

4. In previous work, the authors demonstrated that CSM contractility is essential for gut elongation and morphogenesis, as discussed in paragraph 3 of the Discussion. In the present study, the authors showed that the effect of calcium waves on intestinal differentiation is not mediated by mechanobiological pathways related to smooth muscle contractions. However, it seems that an additional control group (ctl+stretch) should be included before ruling out the role of stretch or contraction in intestinal differentiation.

As we initially received the reviewer comments, we tried hard adding the ctl+stretch group, without success because of systematic contaminations of all 4 experiments we additionally set up. These are performed outside of the incubator (motor cannot be placed inside), in a non-sterile environment, and although we had success with 2 such experiments (presented in the first version of the manuscript in Fig.4), we are unfortunately not managing to reproduce these without contamination. We however do not believe that stretch should anyhow perturb differentiation in non-nifedipine treated guts – if anything it should enhance it. We have also down-toned the interpretation of this experiment in the discussion:

“Finally, our mechanical stimulation experiment does not completely rule out the possibility that LSM differentiation is induced by CSM contractions. The cyclic stretch we applied was sufficient to significantly elongate the guts (compared to non-stretched controls in the same bath), but we cannot exclude that the amplitude (5-10% at 1 cpm, with re-stretching of the preparation twice per day) may have been insufficient to continuously stimulate LSM differentiation.”

5. The choice of experimental models appears inconsistent, with mice used for the initial investigation of calcium waves and mechanical stimulation on intestinal smooth muscle differentiation (Figures 1-4), followed by a switch to chicken embryos (Figures 5a-e), and a return to mice (Figures 5f-h). Additionally, the authors should specify the method for quantifying immunostaining in Figures 5f and 5g. The current images in these panels are blurry, making it challenging to ascertain whether the dots indicated by arrows represent individual cells or cell clusters.

We have now made the manuscript more homogeneous, presenting only mouse data.

We now clarify the method of quantification of the immunostainings (new Figure 6) in the Materials and Methods:

“We quantified the surface density of KIT positive cells by drawing a region-of-interest around the gut tunica muscularis (LSM + ENS + CSM), visually counting the number of KIT+ cells in this ROI, and dividing the count by the area of the ROI”

We have reproduced the results of this experiment in additional 2 controls and 3 nifedipine samples, for a total of now $n=7$ controls and $n=8$ nifedipine (Fig.6g-i). We now provide high magnification images along with DAPI staining showing the emergence of the KIT+ population at the gut periphery instead of LSM in nifedipine conditions, which is one of the key new results of this report.

6. In Figures 5a to 5d, the part in parentheses "(relative to control)" in the y-axis title should be revised to "(relative to GAPDH)" or "(relative to RPLPO)" for clarity and accuracy.

We do not present the chicken PCR data anymore for the reasons stated in the introduction.

Reviewer #2 (Remarks to the Author):

In this manuscript Chevalier et al., analyzed in mouse and chicken embryo the effect of calcium waves to smooth muscle differentiation. The authors applied multiple organ cultures combined with embryonic gut motility analysis, calcium imaging for smooth muscle and enteric neurons, and immunocytochemistry to characterize the development of smooth muscle and motility before birth/hatching. Greater understanding of the intestinal motility and emphasizing the complex molecular and mechanobiological pathways preceding the visceral smooth muscle differentiation are a great value to both congenital gastrointestinal diseases and muscle biology field. The presented findings provide mechanistic insight into how calcium waves are required for the smooth muscle differentiation in the embryonic gut.

Comments and recommendations:

1) last paragraph of the Introduction section is highly similar to the Abstract.

We modified in the revised version both the abstract and this last paragraph.

2) Introduction, first paragraph: muscularis mucosae “is located between lamina propria and submucosa layers” instead of “under the epithelium”

3) Introduction, fourth paragraph: *Drosophila melanogaster* instead *D.Melanogaster*

We corrected this.

4) Results (first paragraph + Fig1). According to previous papers smooth muscle differentiation starts around E11.5 in the mouse intestine (Sharbati, 1982, J Anat; MCHugh, 1995). This contradicts the authors' statement about smooth muscle development. More specifically, according to McHugh, 1995 (paper also cited in this manuscript), alpha-smooth muscle isoactin gene expression is first evident in the E11.5 hindgut. In contrast of cranio-caudal differentiation pattern of the embryo the intestinal smooth muscle differentiation initiates from both caudal and cranial end of the gastrointestinal tract. This should be discussed.

5) To clarify the controversy, I would recommend for the authors to include to Fig 1 additional images from 12.5 dpc midgut and 13.5 dpc hindgut cross-sections immunostained with alpha-smooth muscle actin antibody.

Our new results Fig.1&2 clarify this, and our new results are consistent with the literature as we now detail in the discussion:

“Our chronology is in agreement with earlier, coarser (both temporally and spatially) investigations of α -SMA isoactin expression by in-situ hybridization (McHugh, 1995): this investigator found low to undetectable α -SMA at E11 in the midgut, but detectable levels at E13; the E11 colon was α -SMA negative at E11, and had moderate levels at E13; the LSM was found to differentiate 2-3 days after the CSM. Our results on LSM differentiation are consistent with a more recent study in the jejunum (Walton et al., 2016): these investigators found no LSM at E14, scattered LSM cells at E14.5 and E15, and a continuous LSM layer only as from stage E16.”

6) What is the source of calcium waves, it is produced by mesenchymal cells or enteric neurons? The E12.5 (dpc) jejunum analyzed in Video 1 should already contain enteric neural crest derived cells. Do you see calcium waves in the aganglionic segment of the gut? To generate experimental aganglionosis in mice you can culture the E10.5 postumbilical midgut or E11.5 hindgut for 2 days and analyze the presence of calcium waves.

The calcium waves are produced by the mesenchyme and are independent of enteric neurons at these early stages:

- We have previously performed culture of aganglionic chicken colon (see Video S11 in [2]), which does not affect smooth muscle differentiation and the emergence of contractility (i.e., underlying calcium waves)
- Application of tetrodotoxin at early stages in the mouse [4] (E13.5-E15.5) or in the chicken [2,5] (E6-E12) does not affect motility, myogenic contractility.
- In Video S8, we now show an example of neural-crest Ca^{2+} activity, which is qualitatively very different from the mesenchymal electric activity.

We have added the following paragraph in the discussion to clarify this:

“We found that spontaneous Ca^{2+} waves (CWs) were generated by the smooth muscle layer immediately after it differentiated, at E12.5 in the midgut and at E13.5 in the colon. This activity is purely myogenic: it is resistant to tetrodotoxin (Holmberg et al., 2007; Roberts et al., 2010; Chevalier et al., 2017) and appears in cultured aneural chicken hindgut (Chevalier et al., 2017).”

7) ...”the nifedipine treated gut samples were also affected, as Tuj1+ cells were scarce, no LSM differentiation, motility was impeded....How the intestinal proliferation and apoptosis rate changes after nifedipine/nicardipine treatments? To increase the quality of their work the authors should consider about including anti-casp3/Tuj1 and anti-casp3/alpha-SMA double immunos to Fig 2. 8) How the ENS patterning and differentiation is affected by nifedipine/nicardipine treatments?

We have now added combined anti-casp3/Tuj1 and anti-casp3/alpha-SMA stainings (Fig.6 and Fig.S1), and have also performed staining for proliferation (phosphorylated histone 3, Fig.S1).

We find that nifedipine indeed induces a strong apoptosis in the regions of the differentiated CSM in E12.5+2 (Fig.S1) and E14.5+2 (Fig.6) samples, but not in E11.5+2 samples (Fig.5). This apoptosis at E12.5+2 leads to almost complete disappearance of the CSM in E12.5+2 samples, which we had previously incorrectly interpreted as an inhibiting effect of nicardipine on CSM differentiation. Importantly however, we do not find apoptosis in the presumptive E14.5+2 LSM, indicating that the inhibition of LSM differentiation (and its shift to ICC differentiation) by nicardipine is not due to toxicity/apoptosis.

Similarly, we find that nifedipine/nicardipine induces apoptosis (toxicity) on ENS neurons in E12.5+2 and E14.5+2 cultures but not in E11.5+2 cultures. We further find in the new results presented in Fig.3 that L-type calcium channels start being distinctly expressed on smooth muscle and ENS neurons in the time-window E12.5-E13.5. Our results therefore indicate that the $CaV1.2$ blockers induce apoptosis of smooth muscle and neurons only when they start expressing L-type calcium channels.

9) Fig S2f is not explained in Results and Legends sections. Which segment of the chicken gut

is shown on Fig S2a-c? 10) Fig S2f labeling: %"dead" cells.....you mean death? How was this analyzed? Describe the method.

We do not present the chicken PCR data anymore for the reasons stated in the introduction.

11) Clone name for the Tuj should be Tuj1.

We corrected this.

12) Please includes what is the molecule recognized by Tuj1 mAb.

We added this information (β III-tubulin).

13) It is not clear why the Rho/Rock inhibition had to be done?

Smooth muscle contractility is in general believed to depend both on the calmodulin-MLCK pathway and the Rho/ROCK pathway (Deng et al., 2012; Rattan et al., 2010). It was therefore of fundamental interest to understand whether these pathways played a role in early intestinal contractility and/or calcium wave propagation. We found that Y-27632 does not alter the motility, indicating that the Rho/ROCK pathway is dispensable for early intestinal contractility or calcium wave generation. We here clarify that this inhibitor cannot be used to affect contractility, as had been assumed in a recent publication [6].

14) Fig.4b size (labeling, column size) is at least twice bigger than Fig. 4a. You should resize the image and diagram.

We simplified this figure (new Fig.7).

15) Page 8. Chicken experiment: which part of the intestine was cultured for 3 days, and analyzed for ACTA2 and ACTG2 expression?

The whole gut (from duodenum to colon) was used for these experiments.

16) "As expected, at E5, the intestinal mesenchymal is not yet differentiated." ---According to Graham et al 2017, J Anat, E5 chicken midgut is alpha-SMA negative, hindgut is alpha-SMA immunoreactive. This controversy should be mentioned/clarified and discussed.

Yes indeed, and we had actually found the same result as in Graham et al. 2017 in our study in chickens as well [2].

17) Fig.5 f and g. Please show sperate channels for DAPI and SMA, KIT antibodies (blue, green, red) because the c-Kit immunoreactive cells are not seen in Fig. 5g. Also include insets or separate figures from high power pictures to prove the presence of c-KIT ICC cells on this section. The background is very high maybe caused by the anti-mouse c-KIT antibody detected with anti-mouse secondary on mouse sections. I would recommend this antibody: rat anti-c-Kit (ACK2, 1:800, eBioscience, USA) used by <https://link.springer.com/article/10.1007/s00441-019-03120-9#Sec2> to specifically stain the mouse c-KIT+ ICCs

Thank you for the antibody recommendation. We now provide high magnification images

(Fig.6) along with DAPI staining showing the emergence of the KIT+ population at the gut periphery instead of LSM in nifedipine.

Reviewer #3 (Remarks to the Author):

Summary: This study uses GCaMP6f mice to show that calcium waves propagate in mouse intestinal mesenchyme prior to α SMA expression in the smooth muscle layer, and L-type calcium channel may be the main calcium channel responsible. The authors tried to use ML-7 and Y-27632 to delineate the involvement of mechanical force (stretching, myosin contraction) in this calcium propagation, but due to the flaws in some experimental design, the conclusion is not fully supported. However, there are some new findings, and calcium signaling in smooth muscle differentiation is an important topic. So the work is valuable for this journal. Please address the following to improve.

Comments:

1. One of the strengths of this paper is the live imaging of calcium propagation as demonstrated in Fig 1. However, the authors did very little to explore and investigate the effects of perturbing calcium signaling to CSM or LSM differentiation. For example, how do different doses of nif affect smooth muscle pattern formation? How does Nif and BayK treatment affect smooth muscle formation differently? How does Nif/BayK treatment affect the orientation of smooth muscle cells?

The effects of different doses of nifedipine is presented in Figure S1 (previously Fig.2) for the CSM and in Fig.6b for the LSM ; no significant effects were found at 0.1 and 1 μ M, while 10 μ M inhibits LSM formation, and drives apoptotic effects in L-type channel expressing cells (CSM, enteric neurons) after E12.5. We did not observe any effect of nifedipine on orientation of the smooth muscle, unlike the findings of Huycke et al. in the chicken. We clarified these differences in the discussion:

“In chicken, 10 μ M nifedipine treatment of E9+3 chicken guts did not abrogate differentiation, but led to misorientation of the outer (LSM) muscle layer, which oriented circumferentially instead of longitudinally (Huycke et al., 2019). This misorientation could be rescued by applying cyclic longitudinal mechanical stress. The experiments we performed in mice show several differences to these findings in chicken: 1° α -SMA staining was mostly absent in the presumptive LSM region in the presence of nifedipine, i.e. differentiation was affected, 2° we did not observe misorientation of the LSM in mouse guts treated with nifedipine in the few regions where LSM was still present (~10% of gut length), 3° we could not rescue LSM differentiation by cyclic longitudinal stretching in the presence of nifedipine.”

2. What is the L-type calcium channels expressing in the LIX1 progenitor mesenchymal cells? Will the expression change during the differentiation process to CSMs, LSMs, and ICCs?

We now added staged $Ca_v1.2$ IHCs to our report, showing the change in the expression of this channel during CSM differentiation. These channels are expressed exclusively on the CSM and enteric neurons at stage E13.5-E14.5; from this data, we infer that the apoptotic effect of nifedipine takes place after these channels are expressed.

3. Is the pre-CSM and pre-LSM calcium wave also mediated by gap junction as the calcium

waves inducing early contractile events? It could be tested by inhibitor of gap junctions. And if so, which connexins are expressed and functioning here?

We now performed experiments with enoxolone at stage E12.5, and added these in Fig.4f,g. and Video S5. The effect is very similar to the one we had previously reported in chicken: an excitation of waves upon application, followed by their extinction after 10 min. We presume connexin 43 is the main type here (cf e.g. <https://pubmed.ncbi.nlm.nih.gov/11735042/>).

4. Fig 3. One of the most effective doses of ML-7 on frequency is 5 uM, but its effect is not included in the video. Why is 10 uM less effective than 5 uM?

5 uM is not less effective than 10 uM. At 10 uM, after the pattern of desynchronized waves sets in, the waves eventually faded away (whereas their frequency is just reduced at 5 uM). No such effects are seen at 5 uM, where only the frequency of the waves is reduced. We have made this clearer in the text:

“Surprisingly, we found that ML-7, an inhibitor of myosine light-chain kinase (MLCK), profoundly altered upstream calcium handling: it significantly reduced CW frequency at 5 μ M (Fig.4i); at 10 μ M and 50 μ M, the synchronized CWs became uncoordinated. Whereas in controls all cells along the diameter in the field of view would light up simultaneously, $n=2/5$ samples at 10 μ M and $n=5/5$ at 50 μ M showed an erratic propagation, with cells along the diameter not lightning up simultaneously (Fig.4k, Video S6), and a strongly decreased propagation speed of $24 \pm 10 \mu\text{m}/\text{sec}$ ($n=7$) compared to controls ($356 \pm 49 \mu\text{m}/\text{sec}$). After 20 min in ML-7 10 μ M and 50 μ M, CWs vanished in, respectively, $n=2/2$ guts and $n=4/5$ guts (Fig.4h,i).”

5. In figure 4b, to better examine the relationship between mechano-stretch and LSM differentiation, one more group could be added as “stretch and no inhibitor” – to check if stretch itself could influence LSM differentiation - it may go through other mechano-sensation channel but not L-type calcium channels. 7. Fig 4b. Cyclic stretch: a stretch only group is needed to show the experimental setup is done properly by showing an effect on LSM coverage.

As we initially received the reviewer comments, we tried hard adding the ctl+stretch group, without success because of systematic contaminations of all 4 experiments we additionally set up. These are performed outside of the incubator (motor cannot be placed inside), in a non-sterile environment, and although we had success with 2 such experiments (presented in the initial manuscript, now Fig.7), we are unfortunately not managing to reproduce these without contamination. We however do not believe that stretch should anyhow perturb differentiation in non-nifedipine treated guts – if anything it should enhance it. We have also down-toned the interpretation of this experiment in the discussion:

“Finally, our mechanical stimulation experiment does not completely rule out the possibility that LSM differentiation is induced by CSM contractions. The cyclic stretch we applied was sufficient to significantly elongate the guts (compared to non-stretched controls in the same bath), but we cannot exclude that the amplitude (5-10% at 1 cpm, with re-stretching of the preparation twice per day) may have been insufficient to continuously stimulate LSM differentiation.”

6. What is the signaling pathway calcium take to promote LIX1+ cells differentiating to KIT+ ICCs but not smooth muscle lineages? Could the author try inhibitors of potential downstream target of calcium signaling pathway (e.g., calmodulin inhibitors)?

These are important questions that will be investigated in follow-up studies.

8. In figure 5, the name of the figure is better changed to a statement instead of a question. Also, panel f and g seems to be under different scale - a scale bar for panel g should be added.

We changed this and now provide clearer pictures of KIT expression in E14.5+2 nifedipine treated new samples, along with DAPI staining.

9. Relying on a couple markers to claim a change in cell fate is insufficient in the omics era. Some basic bulk-RNAseq should be performed to support and show the overall change in cell fates when calcium is perturbed.

We understand but these are beyond our means currently.

10. Ctl video is missing for Y-27632 treated sample

We added the videos of control, Y-27632 10 uM, and after the increase to 50 uM (new Video S7).

Bibliography

- [1] H. K. Graham, I. Maina, A. M. Goldstein, and N. Nagy, *Intestinal Smooth Muscle Is Required for Patterning the Enteric Nervous System*, *J. Anat.* **230**, 567 (2017).
- [2] N. R. Chevalier, V. Fleury, S. Dufour, V. Proux-Gillardeaux, and A. Asnacios, *Emergence and Development of Gut Motility in the Chicken Embryo*, *PLoS One* **12**, e0172511 (2017).
- [3] R. Amedzrovi-Agbesi, A. El Merhie, N. J. Spencer, T. Heberd, and N. R. Chevalier, *Tetrodotoxin-Resistant Mechano-Sensitivity and L-Type Calcium Channel-Mediated Spontaneous Calcium Activity in Enteric Neurons*, *Exp. Physiol.* **1** (2024).
- [4] R. R. Roberts, M. Ellis, R. M. Gwynne, A. J. Bergner, M. D. Lewis, E. A. Beckett, J. C. Bornstein, and H. M. Young, *The First Intestinal Motility Patterns in Fetal Mice Are Not Mediated by Neurons or Interstitial Cells of Cajal*, *J. Physiol.* **588**, 1153 (2010).
- [5] N. R. Chevalier, N. Dacher, C. Jacques, L. Langlois, C. Guedj, and O. Faklaris, *Embryogenesis of the Peristaltic Reflex*, *J. Physiol.* **597**, 2785 (2019).
- [6] Y. Yang, P. Paivinen, C. Xie, A. L. Krup, T. P. Makela, K. E. Mostov, and J. F. Reiter, *Ciliary Hedgehog Signaling Patterns the Digestive System to Generate Mechanical Forces Driving Elongation*, *Nat. Commun.* **12**, 1 (2021).

We thank the referee for the feedback on our revision, and provide answers below in green to his questions:

1. In several places, the authors claim that Ca²⁺ waves are mediated by Cav1.2. Without genetic manipulation or an isoform-specific L-type Ca²⁺ channel inhibitor, this conclusion seems premature. Nifedipine and Bay K8644 do not act exclusively on Cav1.2. Can the authors rule out the involvement of Cav1.3 or Cav1.1 in these Ca²⁺ waves?

We specify CaV1.2 because we do not know of any reports of CaV1.1 or CaV1.3 in intestinal smooth muscle. CaV1.1 is expressed exclusively in skeletal muscle. CaV1.3 has been shown to be expressed in gut epithelium, not smooth muscle (<https://journals.physiology.org/doi/full/10.1152/ajpgi.00394.2016>). In contrast it is well known that CaV1.2 is fundamental in gut smooth muscle function : <https://faseb.onlinelibrary.wiley.com/doi/pdf/10.1096/fj.05-5292fje>

2. Figures 1 and 2: The authors should use more objective imaging analysis. Data are presented descriptively. How confident are the authors that there is no α -SMA, and therefore no smooth muscle fibers, at E11.5? The images show robust signals in all three parts of the gut at E11.5. What are the negative controls for these immunostains?

We now detail the criteria on which the presence of SMA fibers were evaluated, at the beginning of the results section :

“We distinguished SMA fibers on the following criteria : 1) a fibrous appearance of the staining, 2) its presence as two distinct layers on either side of the epithelium, and not extending all the way to the outer rim of the sample (serosa). Staining which did not fulfill these criteria was considered non-specific antibody labeling.”

3. Figure 5: In the nifedipine group, both α -SMA and Tuj1 were significantly upregulated. Are these signals artifacts? If they are real, the authors should comment on their relevance.

These differences were not systematic and depending on the sample / area chosen the signal could be stronger in one group, or the other. Concerning the ENS aggregates in the nifedipine group, we specified in the legend that “Aggregates of Tuj1 at the mesenteric border could be found in both control and nifedipine samples.”

4. Figure 7: Since the guts were stretched longitudinally, these experiments do not provide convincing evidence to answer the question “Is LSM differentiation driven by CSM contractions?”

As we explain right after the chapter heading, the force component that could affect LSM differentiation and orientation is the cross-strain in the longitudinal direction when the gut is circularly compressed by the CSM contraction. To put it in simple terms, if you compress radially uniformly a piece of playdoh, as the CSM contractions do, the playdoh will elongate because of the cross-stress in the longitudinal direction. We have identified this component clearly in previous publications. In the experiment, we are directly applying a longitudinal stress, which is also much more convenient from a practical point of view.

“CSM contractions induce circumferential compression and also longitudinal tensile stress of similar magnitude (Chevalier et al., 2021b) in the embryonic gut. CSM contractions are inhibited by nifedipine in E14.5 guts, raising the possibility that the inhibition of LSM differentiation by nifedipine at E14.5+2 could be due to the absence of the cyclic, longitudinal tensile mechanical stress associated with CSM contractions.”